# A cost-effective alkaline polysulfide-air redox flow battery enabled by a dual-membrane cell architecture

Yuhua Xia[1,8], Mengzheng Ouyang[1,8✉], Vladimir Yufit[1,2], Rui Tan[3], Anna Regoutz[4], Anqi Wang [3], Wenjie Mao[3], Barun Chakrabarti [1,5], Ashkan Kavei[1,6], Qilei Song [3], Anthony R. Kucernak [6,7] & Nigel P. Brandon[1,6]

With the rapid development of renewable energy harvesting technologies, there is a significant demand for long-duration energy storage technologies that can be deployed at grid scale. In this regard, polysulfide-air redox flow batteries demonstrated great potential. However, the crossover of polysulfide is one significant challenge. Here, we report a stable and cost-effective alkaline-based hybrid polysulfide-air redox flow battery where a dual-membrane-structured flow cell design mitigates the sulfur crossover issue. Moreover, combining manganese/carbon catalysed air electrodes with sulfidised Ni foam polysulfide electrodes, the redox flow battery achieves a maximum power density of 5.8 mW cm$^{-2}$ at 50% state of charge and 55 °C. An average round-trip energy efficiency of 40% is also achieved over 80 cycles at 1 mA cm$^{-2}$. Based on the performance reported, techno-economic analyses suggested that energy and power costs of about 2.5 US$/kWh and 1600 US$/kW, respectively, has be achieved for this type of alkaline polysulfide-air redox flow battery, with significant scope for further reduction.

[1] Department of Earth Science and Engineering, Imperial College London, London SW7 2AZ, UK. [2] Addionics Ltd., Imperial White City Incubator, 80 Wood Lane, London W12 0BZ, UK. [3] Department of Chemical Engineering, Imperial College London, London SW7 2AZ, UK. [4] Department of Chemistry, University College London, 20 Gordon Street, London WC1H 0AJ, UK. [5] WMG, Warwick Electrochemical Engineering Group, Energy Innovation Centre, University of Warwick, Coventry CV4 7AL, UK. [6] RFC Power Ltd., Imperial White City Incubator, 80 Wood Lane, London W12 0BZ, UK. [7] Department of Chemistry, Imperial College London, London SW7 2AZ, UK. [8] These authors contributed equally: Yuhua Xia, Mengzheng Ouyang. ✉email: m.ouyang15@imperial.ac.uk

The rapid deployment of renewable energy such as solar and wind power has driven the development of grid-scale long-duration energy storage technologies[1,2]. Redox flow batteries (RFBs) have received great attention for medium- to large-scale energy storage applications[3,4]. Compared to conventional rechargeable batteries[5], RFBs provide a number of unique benefits, such as modularity, independence of power and energy, tolerance of deep charge and discharge and enhanced safety[6]. A large number of RFB systems have been proposed and demonstrated in recent years, of which the most-developed system today is the all-vanadium RFB (VRFB). However, the current cost of VRFBs ($400–500/kWh) remains too high for widespread commercial adoption[7]. Amongst various redox chemistries, the polysulfide redox couple[8] has attracted great attention due to its low cost, environmentally benign nature and high energy density owing to the high aqueous solubility of polysulfide[9]. For these reasons, the organic-solvent-based polysulfide chemistry has been widely looked into in lithium-sulfur batteries and sodium-sulfur batteries[10,11]. On the other hand, the aqueous polysulfide redox couple has been adopted in a number of RFB systems, including polysulfide-bromine[12–14], polysulphide-iodide[7,15–17], polysulphide-ferrocyanide[18] and polysulfide-air (PSA)[19–21]. Among these, the PSA system has the lowest chemical cost of storage due to its use of the inexpensive oxygen redox couple.

Analysis of the prior art[19] demonstrates the development in 2011 of an all-alkaline PSA RFB system with alkaline-based oxygen(air) redox couple and a polysulfide redox couple, separated by an anion-exchange membrane (AEM). During the discharge process, shorter-chain-length polysulfides are oxidised into longer-chain-length polysulfides in the anolyte while $O_2$ is reduced to $OH^-$ in the catholyte, and vice versa. The reactions of both half-cells and the overall cell reaction are summarised in Eqs. 1–3, based on which the alkaline PSA RFB has a standard equilibrium potential of 0.88 V.

$$\text{Positive side}: O_2 + 2H_2O + 4e^- \rightleftharpoons 4OH^- \; E^0 = 0.401V \text{ vs SHE} \tag{1}$$

$$\text{Negative side}: 2S_2^{2-} \rightleftharpoons S_4^{2-} + 2e^- \; E^0 = -0.476V \text{ vs SHE} \tag{2}$$

$$\text{Cell}: 4S_2^{2-} + O_2 + 2H_2O \rightleftharpoons 2S_4^{2-} + 4OH^- \; E^0 = 0.877V \tag{3}$$

Although this alkaline PSA RFB shows great promise for large-scale energy storage, there are several scientific challenges, such as the development of low-cost catalysts for the polysulfide and oxygen reactions, as well as the crossover of polysulfide through AEMs.

One key challenge of PSA RFBs is the development of low-cost electrocatalysts for both polysulfide oxidation/reduction reactions (PSOR/PSRR) and oxygen evolution/reduction reactions (OER/ORR). The reaction mechanisms of PSOR/PSRR are still not fully understood due to the complex chemistry of aqueous polysulfide solutions. It is commonly believed that polysulfide ions with a chain length between 2 and 5 ($S_x^{2-}$, $x = 2$–5) exist in large quantities within alkaline polysulfide solutions in addition to bisulfide ions ($HS^-$), hydroxide ions ($OH^-$) and alkali-metal cations (e.g., $Li^+$, $Na^+$ and $K^+$)[22,23]. These varieties of sulfur species are in dynamic equilibria with each other within aqueous polysulfide solutions. Transition metal sulphides such as nickel sulphides ($NiS_x$)[13,14] and cobalt sulphides ($CoS/CoS_2$)[7,16] have shown promising performance for polysulfide reactions. While the aqueous polysulfide electrolyte must be alkaline to supress hydrogen sulphide formation[24], the air electrolyte can be alkaline or acidic in PSA systems[20,24,25]. The use of an acidic air electrolyte however requires expensive platinum-group metal (PGM) catalysts and prevents the use of polymer-based cation-exchange

membranes (CEM) due to the neutralisation of two electrolytes in the long-term. In contrast, the use of an alkaline air electrolyte does not suffer from the neutralisation problem and also enables the use of non-precious metals and metal oxides as the air electrode[26], at the cost of a lower open-circuit voltage (OCV) (0.88 V vs 1.705 V for acidic-catholyte-based system).

Another key challenge of polysulfide-based RFBs is the crossover of polysulfides through membranes which leads to self-discharge and rapid capacity decay. Conventionally, a PSA RFB uses single-AEM or single-CEM as the membrane separator[24,27], as shown in Fig. 1a, b. The standard cell design for this alkaline PSA RFB adopted a single-AEM, as shown in Fig. 1a[25]. However, due to the high permeability of polysulfide ions through the AEM, this alkaline PSA RFB suffered from the crossover of sulfur species towards the air-side half-cell, leading to the loss of active materials within the anolyte and poisoning of the sulfur-intolerant air electrode. CEMs, such as Nafion membranes, show good rejection of polysulfides predominantly through Donnan exclusion however the hydroxide conductivity is also poor due to the same effect[28], limiting the OER reactivity of air electrodes. Although significant efforts have been devoted to the synthesis of new membranes with a size-selective function[28], it remains challenging to effectively block aqueous polysulfide ions. For this reason, despite its unique advantages, there have not been any stable alkaline PSA RFB systems reported prior to this work.

In recent work on PSA RFBs, termed air-breathing aqueous sulfur flow batteries[24], Chiang and co-workers demonstrated the operation of the flow battery by using acidic-catholyte ($Li_2SO_4$ in $H_2SO_4$) and alkaline polysulfide anolyte ($Li_2S_2$ in LiOH) separated by a ceramic electrolyte (Lithium Super Ionic Conductor, or LiSICON). PGM catalysts were used for the OER/ORR reactions, such as platinum mesh or dual cathodes coated with $IrO_2$ as OER catalyst and Pt black as the ORR catalyst. However, the PGM catalysts and LiSICON membrane used in their demonstration work are expensive, and the high membrane resistance limits the power density. Low-cost membranes and non-PGM catalysts are still required to further improve the scalability of the PSA flow battery system.

In this work, we demonstrate a stable alkaline PSA flow battery using a modular dual-membrane architecture, by combining an AEM and a CEM in each individual cell. As illustrated in Fig. 1c, the dual-membrane design mitigates the crossover of polysulfide and enables the use of commercially available or synthetic polymer ion-exchange membranes (IEMs). Our modular design reduces membrane resistance and enhances peak power density. Our system was cycled 80 times at 1 mA cm$^{-2}$ displaying 40% round-trip energy efficiency as well as reduced energy and power cost relative to prior work. Furthermore, the innovative dual-membrane design allows the bespoke combination of CEM and AEM membranes as well as integration of polysulfide catalysts and OER/ORR catalysts, providing more flexibility for RFB designs in the future.

## Results
**Characterisation of the Ni foam electrode**. During the operation of the PSA RFB, the ORR/OER reaction happens at the air electrode, while PSOR/PSRR occur at the polysulfide electrode, as shown in Fig. 2a. In this work, the air electrode of the PSA RFB is carbon-supported manganese dioxide ($MnO_2/C$), comprising $MnO_2$ particles (~50 nm) and conductive carbon particles with similar particle size on top of a Ni mesh current collector, as shown in the SEM images in Fig. 2b, c. Ni foam was adopted as the aqueous polysulfide electrode due to its low cost, intrinsic porosity, and high catalytic activity towards PSOR/PSRR[13,15,29].

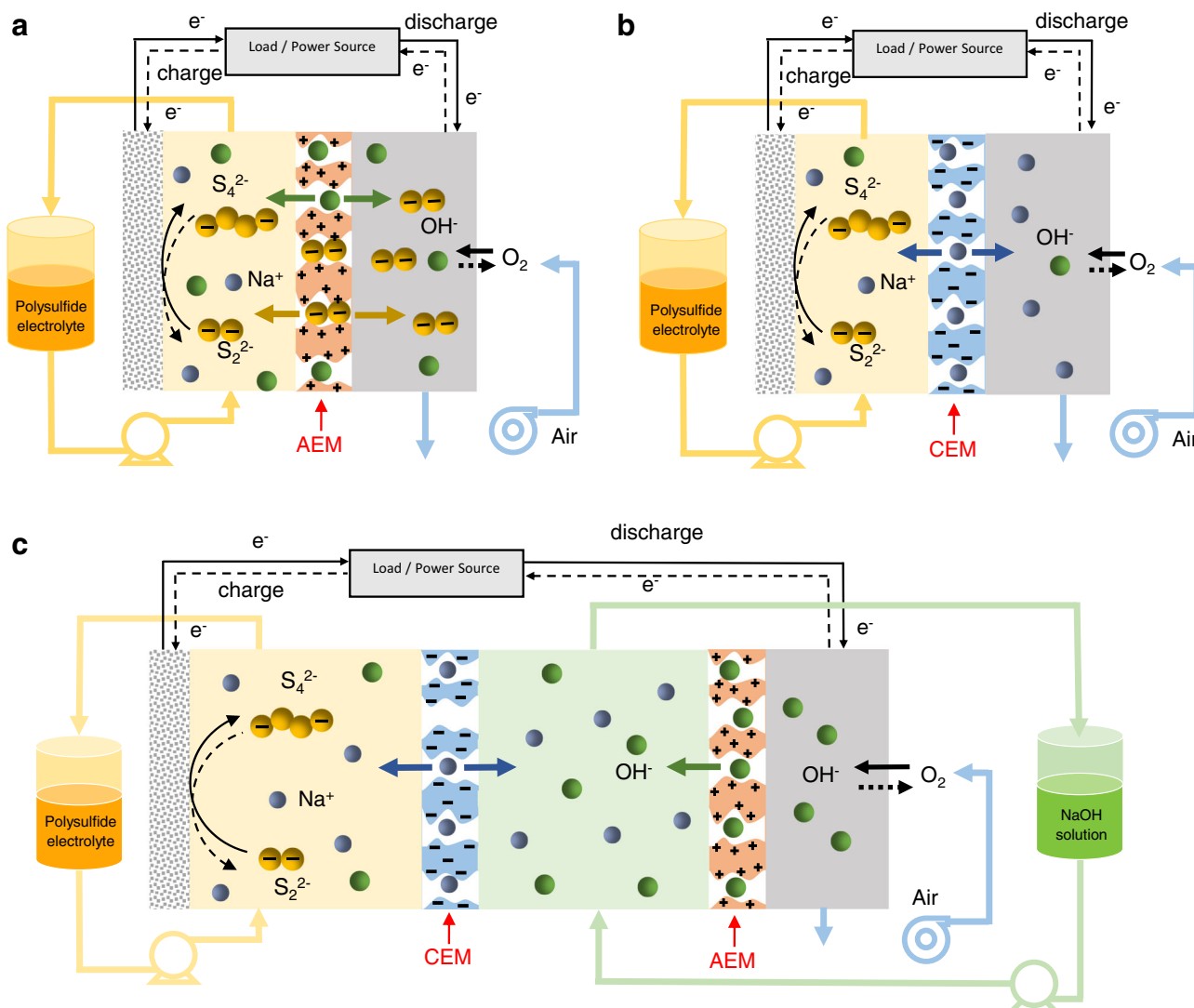

**Fig. 1 Schematic diagrams of alkaline polysulfide/air redox flow battery systems. a** RFB based on single anion-exchange membrane. **b** RFB based on single cation-exchange membrane. **c** RFB based on double-membrane design combining anion-exchange membrane and cation-exchange membrane.

In order to improve its catalytic activity, the Ni foam electrode was pre-treated successively in hydrochloric acid and aqueous polysulfide electrolyte in order to convert the surface Ni into $NiS_x$[29]. Figure 2d shows the photo of as-received nickel foam. SEM images in Fig. 2e, f show that the as-received Ni foam has a smooth surface. After pre-treatment in acid and aqueous poly-sulfide solution, the Ni foam turns black, as shown in Fig. 2g, and the surface of the Ni foam was covered with a layer of nano-particles, as shown in Fig. 2h, i. EDX measurements in Supplementary Fig. 1 indicate that S and Ni were the two major elements existing on the surface of the pre-treated Ni foam, suggesting that the surface species were likely to be $NiS_x$ compounds. XRD patterns (Supplementary Fig. 2) of the pre-treated Ni foam however were identical to that of the as-received Ni foam, where the three observed peaks are attributed to Ni. This suggests that the particles formed on the pre-treated Ni foam surface are amorphous $NiS_x$ compounds. XPS analysis (Fig. 2j, k) suggests that the $NiS_x$ species formed on the pre-treated Ni foam surface was predominantly NiS, with a small contribution from $Ni_3S_2$. Detailed discussion of the XPS measurement for the pre-treated Ni foam can be found in the supplementary note 1 XPS spectroscopy on sulfidised nickel foam, and in Supplementary Fig. 3.

To investigate the catalytic activity of the as-received and sulfidised Ni foam electrodes towards PSRR/PSOR, polarisation measurements were carried out in a $1\,M\,Na_2S_2 + 1\,M\,NaOH$ electrolyte using the three-electrode cell setup, as shown in Supplementary Fig. 4. Results of the polarisation measurements are demonstrated in the form of Tafel plots, as shown in Fig. 2l. The exchange current densities ($i_0$) and the Tafel slopes (b) of these two nickel foam electrodes are extrapolated from the Tafel plots and summarised in Supplementary Table 1. The as-received nickel foam electrode exhibited an asymmetrical feature towards the cathodic and anodic polarisation, where the anodic polarisation current was much higher under the same overpotential. In addition, its anodic Tafel slope ($252\,mV\,decade^{-1}$) was also evidently higher than the cathodic Tafel slope ($196\,mV\,decade^{-1}$). This indicates that the electro-oxidation and reduction of aqueous polysulfides using nickel as the catalyst follows different mechanisms, consistent with observation in previous studies[29]. An average geometric exchange current density of $2.57\,mA\,cm^{-2}$ was achieved for the as-received Ni foam electrode. In comparison, the sulfidised nickel foam electrode showed a significantly improved catalytic activity, achieving an average geometric exchange current density of $6.81\,mA\,cm^{-2}$, 2.65 times that of the as-received nickel foam electrode. It also exhibited a much more reversible performance

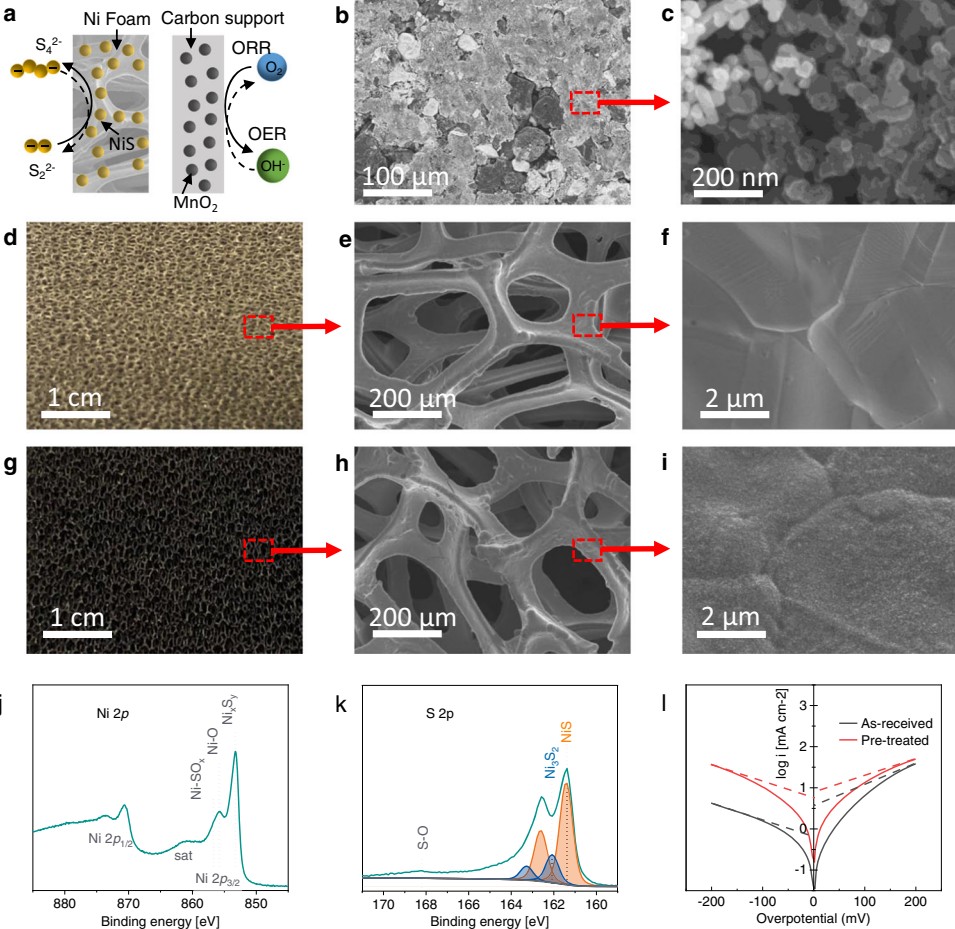

**Fig. 2 Characterisation of electrodes. a** Diagram of sulfidized Ni foam electrode and MnO2-based air electrode for OER and ORR reactions. **b**, **c** SEM images of the commercial MnO2-based air electrode. **d** Photo of the as-received Ni foam. **e**, **f** SEM images of the as-received Ni foam. **g** Photo of the sulfidised Ni foam. **h**, **i** SEM images of the sulfidised Ni foam. **j**, **k** XPS core level spectra of the sulfidised Ni foam. **l** Polarisation measurements of the as-received and sulfidised Ni foams.

towards the electro-oxidation and reduction of aqueous polysulfide electrolyte, in view of the similar cathodic and anodic exchange current densities and Tafel slopes shown in Supplementary Table 1 and Fig. 2l.

The overall performance enhancement of sulfidised nickel can be explained by the enhanced electron transfer from the strongly coupled $Ni/NiS_x$ interface[30,31] and the much faster polysulfide adsorption/desorption rate from the transient metal sulphide surface compared with the partly sulfidised metal surface[32]. The more significant improvement on the cathodic branch is consistent with previous observations[33]. This indicates $NiS_x$ plays a vital role in the electrochemical reduction of polysulfide species. However, detailed analysis of this phenomena merits in-depth study in the future.

**Diffusivity determination of aqueous polysulfide ions through ion-exchange-membranes**. To study the diffusivity of polysulfide ions through the membranes, permeation experiments were performed using an H-cell setup for single-membrane structure and RFB setup for dual-membrane structure, as shown in the schematics in Fig. 3a. Photographic pictures of both setups are shown in Supplementary Fig. 5. The schematics of both setups' components are shown in Supplementary Fig. 6. The concentration of polysulfide was determined by UV-Vis spectroscopy measurements after each given time. When testing a single-membrane structure, two commercially available IEMs were used, namely Nafion 117

CEM and FAA-3-PK-130 AEM, which are also the IEMs used in the actual PSA RFBs. The UV-Vis absorbance was first calibrated to the concentration of $Na_2S_2$ solution, the calibration spectra are shown in Supplementary Fig. 7a, which shows that dilute polysulfide solutions have two characteristic absorption peaks, at around 304 nm and 369 nm. However, due to the stronger absorption at 304 nm, the absorbance of $Na_2S_2$ solutions with a concentration above 5 mM exceeded the detection limit of the UV-Vis spectrometer. Therefore the 369 nm peak was used to determine the amount of permeated polysulfide ions. However, unlike single-solute solutions, of which the UV-Vis absorbance is generally acknowledged to be proportional to its concentration, the composition of aqueous polysulfide solutions is complex with various polysulfide ions in dynamic equilibria with each other, and the concentration distribution of various polysulfide ions is affected by a number of factors such as concentration, pH and temperature[34]. Nevertheless, in this work, the absorption peak intensity of aqueous polysulfide solution (<10 mM $Na_2S_2$) was found proportional to its concentration, as demonstrated in Supplementary Fig. 7a. The molar attenuation coefficient of polysulfide ions at 369 nm was calculated to be 187 $dm^3\ mol^{-1}\ cm^{-1}$.

The UV-vis absorption spectra of the permeate solution in single-membrane and dual-membrane structure after various times are shown in Supplementary Fig. 7b–e. The absorbance of the permeate solution was proportional to the permeation time, the result is shown as cumulative flux in Fig. 3b. The cumulative

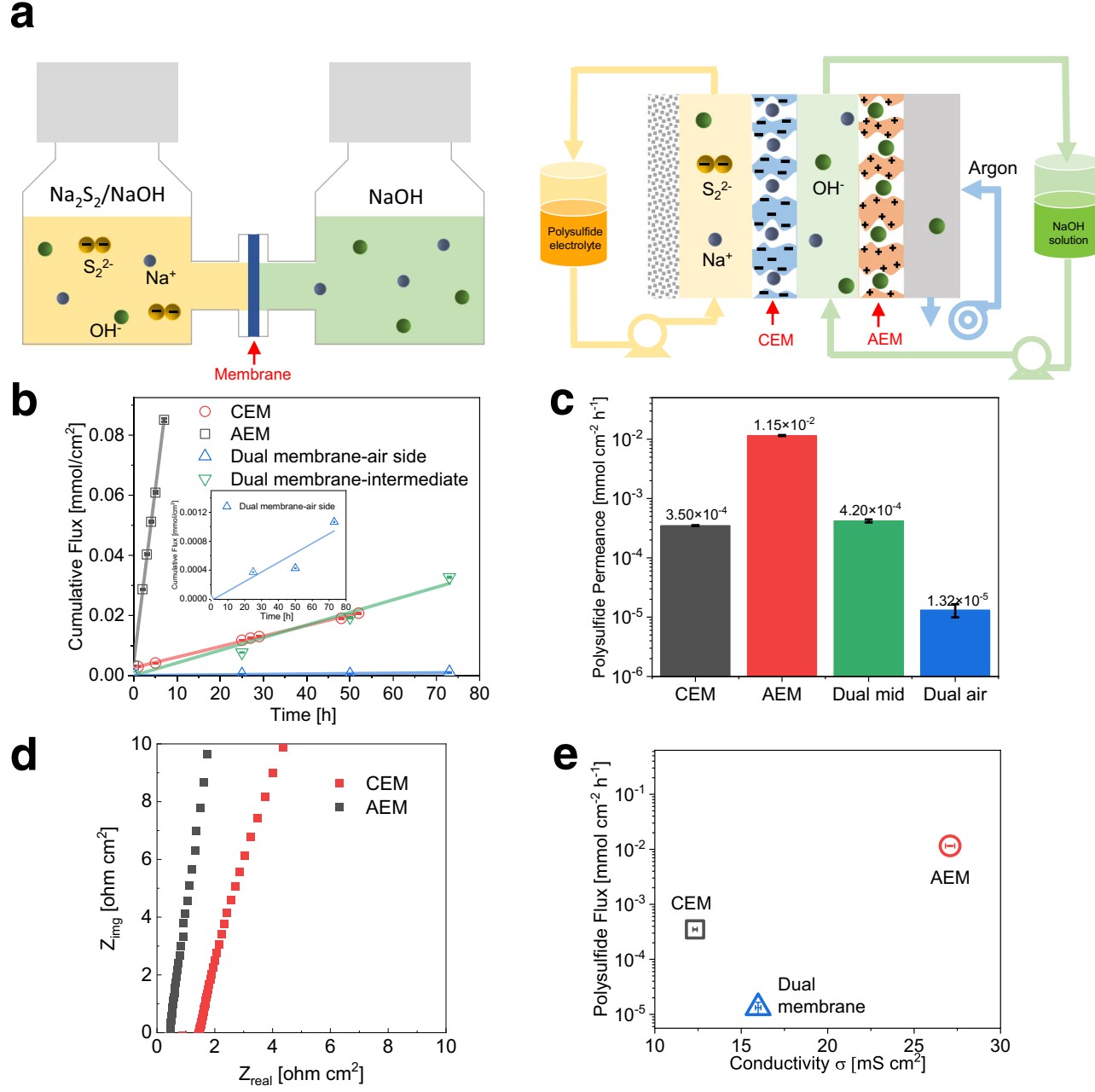

**Fig. 3 Diffusion of polysulfide species through ion-exchange membranes. a** Left: H-cell setup for polysulfide crossover determination on single-AEM (FAA-3-PK-130) and CEM (Nafion 117); Right: redox flow battery setup for polysulfide crossover determination on the dual-membrane structure. **b** Cumulative crossover of the permeate solution in single-AEM, single-CEM and dual-membrane setup against permeation time, inset shows the dual-membrane polysulfide crossover measured at the air side. **c** Calculated polysulfide flux across single-AEM, single-CEM and dual-membrane structure. **d** EIS of CEM and AEM. **e** The polysulfide flux versus membrane conductivity. The error bars show the UV-Vis measurement error, conductivity measurement error and fitting errors.

flux is the total amount of species transported across the membrane normalised by the effective membrane permeation area. The straight line fitted against permeation time indicates a constant flux of polysulfide ions through the IEMs. The calculated flux of polysulfide ions through Nafion 117 CEM, FAA-3-PK-130 AEM are $3.50 \times 10^{-4}$ and $1.15 \times 10^{-2}$ cm$^2$ s$^{-1}$ (Fig. 3c), respectively, indicating a 32.9 times lower transportation rate of polysulfide ions through Nafion 117 CEM than FAA-3-PK-130 AEM. This is also demonstrated by the evidently slower colour deepening of the permeate solution over time in the H-cell setup assembled with Nafion 117 CEM. To conclude, FAA-3-PK-130

AEM would allow OH$^-$ to pass through but is more vulnerable towards polysulfide crossover, while Nafion 117 CEM would allow Na$^+$ to pass and have a higher resistance of polysulfide crossover but it would block OH$^-$.

The ohmic resistance of FAA-3-PK-130 AEM and Nafion 117 CEM in 1 M NaOH solution was determined as the intercept of the impedance signal with the Zreal axis in the complex impedance plot (Fig. 3d) obtained via electrochemical impedance spectroscopy (EIS) measurements. The AEM shows 1/3 the ohmic resistance of the CEM ($0.46 \pm 0.01$ vs. $1.44 \pm 0.03$ $\Omega$ cm$^2$), showing that the AEM enables a faster rate of ion transport. The

relation of sulfur crossover rate versus conductivity of AEM and CEM are shown in Fig. 3e. The AEM exhibits 3.1 times conductivity and 32.9 times polysulfide crossover flux compared with the CEM.

**Dual-membrane flow cell design**. The significantly lower diffusion coefficient of aqueous polysulfide ions through Nafion 117 CEM compared to FAA-3-PK-130 AEM suggests that substitution of the AEM by CEM in the flow cell design could be beneficial in mitigating the polysulfide crossover issues of this alkaline PSA RFB. However, concerns of high charge transfer/mass transport losses for the OER arises due to the limited capacity and conductivity of $OH^-$ of the CEM, especially under high operating current densities. To confirm these speculations, a single-cell alkaline PSA RFB with either Nafion 117 CEM or FAA-3-PK-130 AEM was assembled and tested in this study. To test the polysulfide crossover through the dual-membrane structure, a dual-membrane RFB was assembled in the glovebox, as shown in Fig. 3a. Three peristaltic pumps were used to circulate 50 mL 0.1 M $Na_2S_2$/1 M NaOH solution to the anode side, 50 mL 1 M NaOH solution to the intermediate plate, and environmental argon to the cathode side respectively to mimic the real PSA RFB operating conditions. All pumping speeds were set to 50 mL min$^{-1}$. Every 25h, polysulfide concentration in the intermediate solution was examined using UV-vis analysis, denoted as Dual-mid. At the same time, the cathode side was flushed with 50 mL of 1 M NaOH for 2 min, and the polysulfide crossover amount on the cathode side was determined by examining the polysulfide concentration of this flushed solution, denoted as Dual-air. The increase in the total amount of polysulfide in the permeate solution normalised by area with time is shown in Fig. 3b, and the polysulfide flux through the dual-membrane structure is shown in Fig. 3c. The amount of polysulfide crossover in the intermediate solution of the dual-membrane structure is identical to that in permeate solution of the single-CEM structure, suggesting the crossover experiments using different setup (H-cell vs RFB) are comparable and both are valid. On the other hand, the polysulfide crossover rate through the dual-membrane structure is much lower than that of single-CEM; the calculated flux (Fig. 3c) is 4% of the CEM and 0.1% of the AEM. This suggests that the dual-membrane structure is effective in blocking polysulfide crossover, and that the AEM can be used to block small concentrations of polysulfide ions.

The effectiveness of the dual-membrane structure was further examined in actual PSA RFB operation. The photographic pictures of the setup are shown in Fig. 4a, b. All full battery cycling in this work was performed using 100% theoretical capacity of $S_2^{2-}$/$S_4^{2-}$ (2.68 Ah L$^{-1}$ for 0.1 M $Na_2S_2$/1 M NaOH anolyte) unless the battery reached the cutoff voltage before achieving the theoretical capacity. High charging potential would severely damage the FAA-3-PK-130 AEM, hence an upper cutoff voltage of 1.4 V was set. Charge and discharge cycling at 1 mA cm$^{-2}$ for the single-AEM-based PSA RFB exhibited a capacity degradation as high as 86% after only 10 cycles, as demonstrated in Fig. 4c. This is deduced to be mainly caused by the loss of sulfur species within the anolyte, which diffused through the AEM to the air-side half-cell. This agrees well with the result of polysulfide diffusivity determination experiment, which shows that polysulfide ions can diffuse 4.6 times faster than in CEM. This was also confirmed by visual observation of the experimental setup (Supplementary Fig. 8), where a considerable amount of orange-coloured solution was found at the outlet of the air-side half-cell after a few cycles.

For the single-CEM-based PSA RFB, the upper cutoff voltage during the electrochemical measurements can be increased due to the intrinsically higher stability of the perfluorinated Nafion membrane. No capacity degradation was observed over 10 charge

and discharge cycles, as demonstrated in Fig. 4d. On the other hand, no yellow/orange-coloured solution was observed at the air-side half-cell during the electrochemical measurements of the single-CEM-based PSA RFB. This was attributed to the low diffusivity of polysulfide ions through Nafion 117 CEM. Compared with the single-AEM-based PSA RFB, the single-CEM-based PSA RFB shows smaller overpotential during cycling, evidenced by its lower charging and higher discharging plateaus. This resulted in a higher voltage efficiency. However, Fig. 4d shows that the charging voltage profile of the single-CEM-based PSA RFB shows a continuing upwards trend instead of a plateau, which becomes more pronounced when using a more concentrated polysulfide electrolyte. This is due to the insufficient supply of $OH^-$ to the air-side half-cell in a longer charging process. The high charging plateau also accelerates $MnO_2$ dissolution[35]. The water turned purple when the cell was disassembled and washed after 20 cycles, indicating the dissolution of $MnO_2$. The detailed cycling voltage profile of single-AEM and single-CEM structured alkaline PSA RFB are shown in Supplementary Figs. 9 and 10.

Since the use of neither CEM nor AEM would allow a standard single-membrane-based PSA RFB to operate steadily, a different dual-membrane-structured cell design was initiated for this alkaline PSA RFB; its principle is shown schematically in Fig. 1c and the exploded view of its compartment is shown Supplementary Fig. 6a. Figure 4b shows the photographic pictures of the dual-membrane-structured cell. Instead of using a single-AEM or CEM, a CEM was placed beside the anode and an AEM was placed beside the cathode. The two membranes were separated by an insulating porous interlayer frame, wherein 1 M NaOH solution was circulated. During the operation of this dual-membrane-structured PSA RFB, the CEM would allow Na$^+$ to pass through but stop the sulfur species (e.g., HS$^-$ and S$_x^{2-}$), so that the sulfur species remained within the polysulfide electrolyte. The AEM allows the $OH^-$ ions to pass through and participate in the OER/ORR reactions on the air-side half-cell. For the minor amount of polysulfide that crosses through the Nafion, this would be flushed away and diluted by the circulating intermediate solution, thereby reducing the amount of polysulfide that can reach the air-electrode-side.

As demonstrated in the cycling curves of the dual-membrane PSA RFB in Fig. 4e, the dual-membrane cell structure was successful in mitigating the crossover of polysulfide ions towards the airside half-cell. It also prevented high charging overpotentials of the CEM-based single-membrane PSA RFB, as demonstrated in Fig. 4e. As a result, the dual-membrane-structured PSA RFB exhibited both higher voltage efficiency and high energy efficiencies, as shown in Fig. 4f–h. In addition, the dual-membrane-structured PSA RFB showed a similar average discharging voltage to the single-CEM-based PSA RFB, where in both setups a Nafion 117 CEM was placed by the anode compartment. This indicates that the observed higher average discharging voltage for the single-CEM-based and dual-membrane-structured PSA RFBs compared to the single-AEM-based PSA RFB is related to the use of Nafion 117 CEM instead of FAA-3-PK-130 AEM. The performance of the dual-membrane-structured alkaline PSA RFB is presented in more detail in the next section.

**Performance of the dual-membrane-structured PSA RFB**. Although operating temperatures up to 55 °C have been reported for aqueous polysulfide-based RFBs in the past[24], it is worth noting that higher operating temperatures will not only improve the kinetics of PSOR/PSRR and OER/ORR but will also accelate the side reactions that lead to the formation of oxysulfur anions within the aqueous polysulfide electrolyte and hence capacity

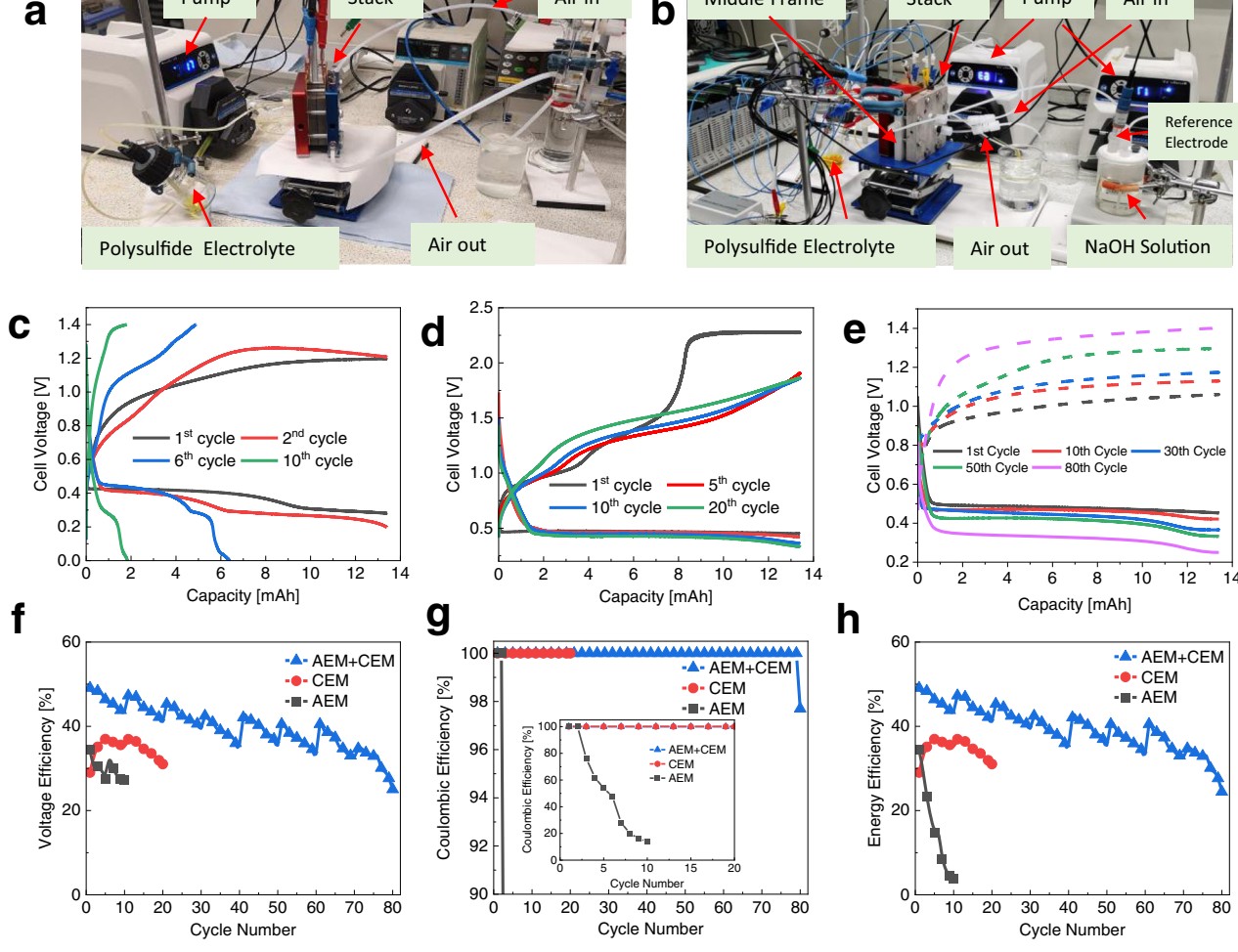

**Fig. 4 Charge and discharge cycling of the alkaline PSA RFB. a** Experimental setup of single-membrane-structured PSA RFB. **b** Experimental setup of dual-membrane-structured PSA RFB. **c** Voltage profile of the PSA RFB with FAA-3-PK-130 AEM over 10 cycles. **d** Voltage profile of the PSA RFB with Nafion 117 CEM over 20 cycles. **e** Voltage profile of the PSA RFB with both FAA-3-PK-130 AEM and Nafion 117 CEM over 80 cycles. **f** Voltage efficiencies against cycle number of the PSA RFBs with either single or dual membranes. **g** Coulombic efficiencies against cycle number of the PSA RFBs with either single or dual membranes. (inset figure shows the minified scale for showing the Coulombic efficiency degradation of single-AEM cell). **h** Round-trip energy efficiencies against cycle number of the PSA RFBs with either single or dual membranes.

degradation[36]. Therefore, electrochemical measurements of this alkaline PSA RFB were mainly conducted at 25 °C ± 0.2 °C in this study as before. Since fresh air was continuously supplied to the air-side half-cell with a mass flow controller, and the pH change of the NaOH intermediate solution can be neglected because of the large tank size used relative to the polysulfide electrolyte, the state-of-charge (SOC) of this alkaline PSA RFB entirely depends on the aqueous polysulfide electrolyte. Previous studies showed that confining the nominal composition of the aqueous poly-sulfide electrolyte between disulphide ($S_2^{2-}$) and tetrasulfide ($S_4^{2-}$) was important in maintaining the stability of aqueous polysulfide electrolytes[24]. Therefore, in this work, the 0% SOC and 100% SOC of the PSA RFB were set as equivalent to the nominal anolyte composition of $Na_2S_4$ and $Na_2S_2$, respectively.

Upon assembly, the alkaline PSA RFB exhibited an OCV of ~0.81 V. EIS measurement of the full cell showed only one depressed semi-circle (correlating to the overall charge transfer process within the PSA RFB), with an intercept on the $Z_{real}$ axis of 2.16 $\Omega\,cm^2$ as demonstrated in Supplementary Fig. 11, which is attributed to the overall ohmic resistance of the full cell including the ohmic resistance of cell components and the contact resistance. It is difficult to distinguish the charge transfer process

of PSOR/PSRR and OER/ORR by the EIS method. However, by inserting a Hg|HgO reference electrode into the intermediate NaOH solution (shown in Fig. 4b), we are able to decouple and monitor the half-cell potentials of the PSA RFB alongside the electrochemical measurements. Under small current densities (<10 mA cm$^{-2}$), where mass transport does not play an important role in the battery performance, the overpotential of each half-cell represents the combination of activation losses and charge transfer losses. Figure 5a shows the overpotential of the air-side and polysulfide-side half-cells during the polarisation curve measurement of the alkaline PSA RFB at 50% SOC and 25 °C. The overpotential of the half-cell reactions follows the trend of: OER>ORR>PSRR>PSOR. On the polysulfide side, the larger overpotential of PSRR compared to PSOR at the same current density is in accordance with the findings of previous studies[13,29]. On the air side, the overpotential of ORR is significantly smaller than that of the OER, which is expected as $MnO_2$ has a better ORR activity compared with its OER activity[37]. Overall, the air-side half-cell showed higher over-potentials compared to the polysulfide-side across all current densities and under all temperatures amongst the polarisation curve measurements. Therefore, it is safe to say that the air-side

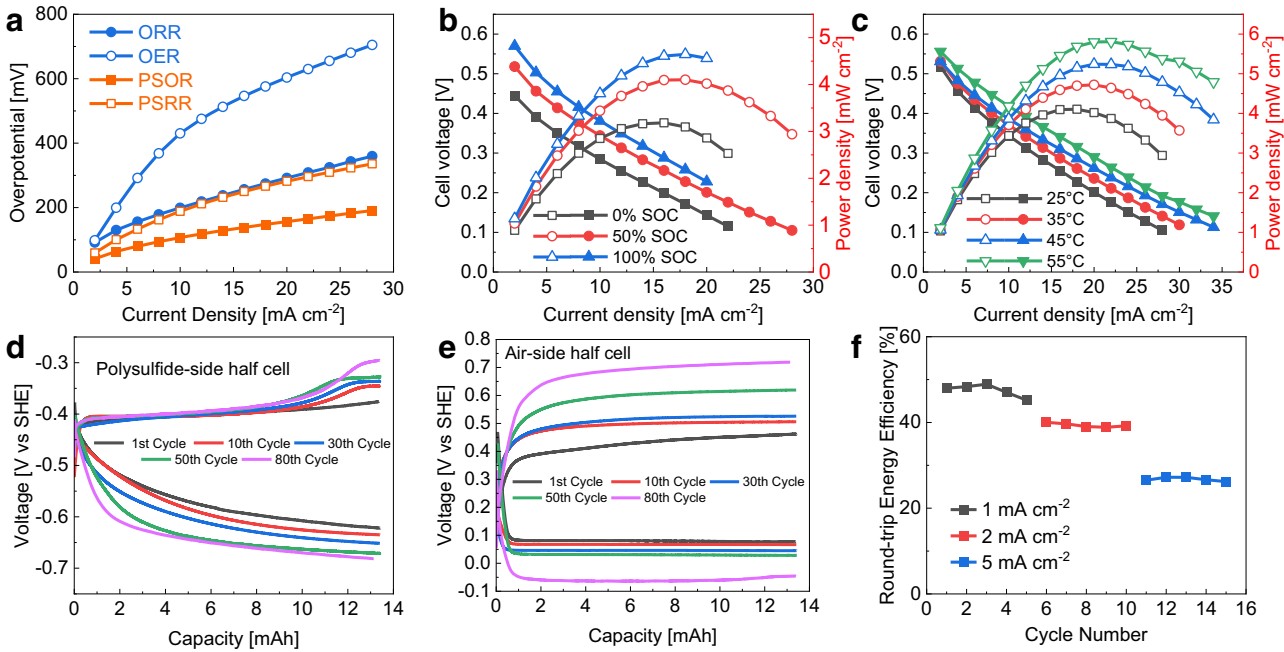

**Fig. 5 Polarisation curve measurement and cycling performance of the alkaline PSA RFB with dual membranes. a** Polarisation curve and power density curve at 25 °C and various SOc. **b** Polarisation curve and power density curve at 50% SOC and various temperatures. **c** Measured half-cell overpotentials against current density at 25 °C and 50% SOc. **d** measured polysulfide-side half-cell potential profile against capacity. **e** Measured air-side half-cell potential profile against capacity. **f** Round-trip energy efficiency against current density.

reaction is the rate-limiting step of the overall battery reaction, highlighting that the development of a better reversible air-electrode is an important future objective.

To investigate the effect of temperature and SOC dependence on the battery peak power density, polarisation curve measurements were conducted at various operating temperatures and SOCs, as demonstrated in Fig. 5b, c. Figure 5b shows that the discharge performance of the PSA RFB was enhanced with the increase of battery SOC. At 25 °C, the peak power densities achieved at 0%, 50% and 100% SOC were 3.18, 4.10 and 4.64 mW cm$^{-2}$, respectively. Since the SOC of the battery is solely determined by the composition of the aqueous polysulfide electrolyte, and given a fixed overall sulfur concentration, the rate of PSOR increases with the average chain length of polysulfide ions. Figure 5c shows the polarisation curves and power density curves of the alkaline PSA RFB at 50% SOC and in the temperature range of 25–55 °C. As expected, higher temperature is beneficial to battery performance. At 50% SOC, a peak power density of 4.10, 4.72, 5.24 and 5.81 mW cm$^{-2}$ was achieved for 25, 35, 45 and 55 °C, respectively. This all-alkaline PSA RFB achieved a 14% higher maximum power density at 55 °C compared with the previously reported state-of-the-art PSA RFB adopting acidic catholyte, and expensive solid-state electrolyte (SSE) separator and precious metal catalyst-based air electrodes[24].

Charge and discharge cycling of this alkaline PSA RFB was performed using a 0.1 M Na$_2$S$_2$ + 1 M NaOH anolyte in order to decrease the experimental duration due to the low operating current density (1–5 mA cm$^{-2}$). A slightly lower OCV of 0.745 V was observed due to the use of a less concentrated polysulfide electrolyte. To exclude the effect of polysulfide oxidation by atmospheric air (as this was not a sealed system), the polysulfide electrolyte was replenished every 10 cycles. The full cycling voltage profile of dual-membrane structured alkaline PSA RFB is shown in Supplementary Fig. 12.

The voltage profile of the full-cell and both half-cells for this alkaline PSA RFB over 80 cycles are presented in Figs. 4e and 5d, e. The charging voltage reached the cutoff voltage of 1.4 V at the 80th cycle and cycling was stopped. Based on the voltage profile, the average discharge and charge potentials of the full-cell and both half-cells during the 1st and 80th cycle are summarised in Supplementary Table 2. The average potential difference between the charge and discharge processes of the air-side reaction (0.322 V during the 1st cycle and 0.634 V during the 80th cycle) was much higher than that of the polysulfide-side reaction (0.183 V during the 1st cycle and 0.236 V during the 80th cycle). This confirmed that the air side is the performance limiting side of this alkaline PSA RFB. Supplementary Table 2 also shows that, over 80 cycles, the average discharge voltage of the full-cell decreased from 0.487 V to 0.327 V (0.16 V in difference) and the average charge voltage of the full-cell increased from 0.992 V to 1.305 V (0.313 V in difference). This indicates that the deterioration of the alkaline PSA RFB's energy efficiency was mainly associated with the charging process. In addition, the charging potential of the air side increased from 0.410 V to 0.688 V, which accounted for 89% of the full-cell potential increase (from 0.992 V to 1.305 V) in the charging process. To conclude, the degradation of this alkaline PSA RFB was mainly attributed to the catalytic activity loss of the air electrode towards OER, which is likely due to the dissolution of MnO$_2$. The MnO$_4^-$/MnO$_2$ redox couple has been investigated in both acidic and alkaline conditions[38,39]. Its standard potential in the alkaline conditions is 0.56 V vs. SHE, which is close to the half-cell voltage of air electrode during charge. Therefore, the oxidation of MnO$_2$ to highly dissolvable MnO$_4^-$ is deduced to be the parasitic reaction that causes the air electrode degradation. The detailed mechanism of air electrode degradation is the focus of future work.

The upper or the lower cutoff voltages were not reached during battery cycling until the 80th cycle, as shown in Fig. 4e. Hence, the capacity and the coulombic efficiency of this alkaline PSA

RFB were both maintained at 100% throughout the 79 cycles, and were 98% for the last cycle. Therefore, the round-trip energy efficiency of this alkaline PSA RFB was calculated to be the multiplication of its voltage efficiency and coulombic efficiency. Figure 4h presents the round-trip energy efficiencies of this alkaline PSA RFB as a function of the cycle number. The repeating pattern on the efficiency figure is attributed to replenishing the polysulfide anolyte every 10 cycles to eliminate the effect of polysulfide oxidation by leaked air[24]. This alkaline PSA RFB exhibited a round-trip energy efficiency of 49% at the 1st cycle, which gradually decreased to 25% after 80 cycles. The average round-trip energy efficiency over 80 cycles was 40%. Compared with the previously reported state-of-the-art PSA RFB which adopted acidic catholyte and used LiSICON separator and PGM catalyst-based air electrodes[24], our dual-membrane all-alkaline PSA RFB achieved a slightly lower round-trip energy efficiency (40% vs 43%) but at both double the number of cycles (80 vs 40) and over three times higher current density (1 mA cm$^{-2}$ vs 0.325 mA cm$^{-2}$) at ambient temperature.

This alkaline PSA RFB was also cycled at higher current densities (2 and 5 mA cm$^{-2}$) for 5 cycles per current density. As shown in Fig. 5f, no evident degradation of the battery efficiency was observed over the five cycles at either 2 or 5 mA cm$^{-2}$. However, compared with the cycling performance at 1 mA cm$^{-2}$, the average round-trip energy efficiency of this alkaline PSA RFB decreased to 39.4% and 27.4% when cycling at 2 and 5 mA cm$^{-2}$, respectively. The low efficiencies of this alkaline PSA RFB were mainly attributed to the poor catalytic activities of MnO$_2$-based air electrode towards OER.

Nafion 117 and FAA-3 were chosen in this work because these are some of the most commonly used commercial CEMs and AEMs in the energy storage field[17,40]. However, dual-membrane cells using two other commercially available membranes were also assembled and tested; the first using Nafion 212 as CEM and FAA-3 as AEM, the second using Nafion 117 and a newly developed AEM PAP-TP-100[41] (commercially available as PiperION). The EIS measurements and first 10 cycles of cells using different combinations of membranes are shown in Supplementary Fig. 13. The Ohmic resistance of Nafion 212/FAA-3-PK-130 was 4% lower than Nafion 117/FAA-3-PK-130 (2.55 ± 0.06 Ω cm$^2$), while Nafion 117/PAP-TP-100 was 5% higher. These differences are small compared with the total impedance which is over 60 Ω cm$^2$. The polarisation and degradation of the first 10 cycles of different membrane combinations are also similar. This suggests that changing the membranes had little influence on the performance of the dual-membrane PSA RFB. This supports the view that the concept is not restricted to the use of particular membranes, and further validates our conclusion that the current rate-limiting part of this system is the electrode, particularly the air electrode.

Another potential problem associated with the future application of dual-membrane structures in PSA RFBs is the osmotic effect, as identified in other RFB system[42]. This is discussed in detail in supplementary note 2 Osmotic effect and its influences.

**Cost analysis of the PSA RFB system**. Cost analysis, including the energy (chemical) cost of storage (C$_{energy}$) and the power cost of storage (C$_{power}$), was performed for this alkaline PSA RFB and previously reported PSA RFB based on the actual materials used and real performances achieved. The detailed calculation methods, assumptions and breakdown of costs can be found in Supplementary Notes 3 and 4. Figure 6 shows a graphical comparison of the dual-membrane alkaline PSA RFB in this work with the previously reported state-of-art PSA RFB using acidic catholyte (noted as Reference in Fig. 6)[24]. Figure 6a shows the quantitative

comparison of some key performance and cost indicators, in which the radial axis is the times of improvement of this work comparing with referenced work. The PSA RFB in this work achieved similar peak power density and round-trip energy efficiency, and 3 times the stable cycling current compared with the referenced work, while the power cost and energy cost are only 1/5 and 1/4, respectively. Figure 6b shows a comparison of the cost of major components of the PSA RFB in these two works, and the total power cost C$_{power}$. Based on the peak power density achieved during the polarisation measurements and the bulk prices of raw materials, the C$_{power}$ of this alkaline PSA RFB is calculated to be 1635 US$/kW. In comparison, the C$_{power}$ of the referenced RFB is recalculated using the actual achieved peak power density and the present costs of the raw materials used, which demonstrates a value of 8358 US$/kW. The much lower power cost for this PSA RFB compared to the previous study is mainly attributed to the much lower cost of Nafion/FAA-3 membranes and MnO$_2$ air electrode used in this study, compared with the SSE membrane and Pt/IrO$_2$ air electrode in the previous study. The calculation process and assumptions of C$_{power}$ is shown in supplementary note 3 Calculation of the power cost of storage. The breakdown costs used to calculate C$_{power}$ of these two cells can be found in Supplementary Table 3 and Supplementary Table 4.

Figure 6c shows a comparison of chemicals cost and the total energy cost C$_{energy}$ of PSA RFBs in these two studies. The reference work used sodium salt in the cost analysis and testing of the air-electrode, and lithium salt in the actual PSA battery test. Therefore, to correctly correlate the battery performance with the cost of energy, lithium salts were chosen to calculate the energy cost of the reference work. Assuming a total dissolved S content of 5 M (energy density 58 Wh/L), this alkaline PSA RFB is calculated to have a C$_{energy}$ of 2.54 $/kWh. In comparision, the referenced PSA RFB calculates a C$_{energy}$ of ~9.77 US$/kWh, under the same S content (5 M). The difference in C$_{energy}$ comes from the lower cost of chemicals used in this work, expecially the catholyte, which is only 4% the cost of referenced work. Nevertheless, both systems demonstrate that the energy cost of polysulfide/air-based RFB systems is the lowest of all battery technologies, lithium ion batteries are reported to be 271 US$/kWh[43], VRFBs 185 US$/kWh[44] and lead-acid battery 260 US$/kWh[45], for example. The calculation process and assumptions of C$_{energy}$ is shown in supplementary note 4 Calculation of the energy cost of storage. The breakdown costs used to calculate C$_{energy}$ of these two cells can be found in Supplementary Table 5 and Supplementary Table 6.

More detailed key performance indicators of this alkaline PSA RFB and the two previously reported PSA systems are summarised in Supplementary Table 7. The alkaline PSA RFB in this work exhibits the highest performance and the lowest cost. However, in contrast to its low energy cost (C$_{energy}$), the power cost (C$_{power}$) of this alkaline PSA RFB is still too high for commercialisation due to its low power density and associated high electrodes/membrane costs. This suggests that future effort should be focused on improving the catalytic activities of the electrodes, especially on the air side, and identifying improved and lower-cost membranes.

## Discussion

The high performance and low cost of the PSA RFB presented here are attributed to the dual-membrane structure.

If a single-membrane structure is used, it would be difficult to use an all-alkaline electrolyte with a single bifunctional air electrode: as the membrane is sandwiched by two electrodes, if the cathode is flooded with KOH solution, it will have a low oxygen feed during discharge (ORR reaction). If saturated by oxygen, it will lack OH$^-$ reactant during charge (OER reaction) unless an AEM is used as the separator membrane, since in alkaline

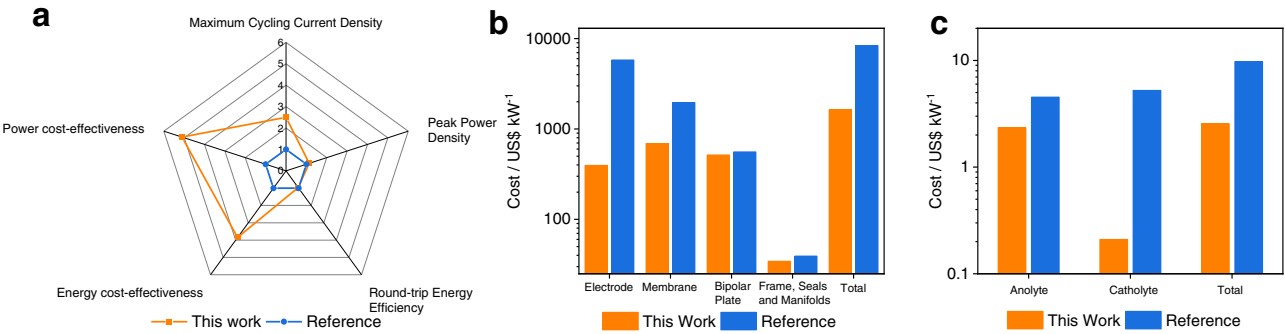

**Fig. 6 Comparison of the performance and costs of polysulfide/air RFBs. a** Key performance indicators (power cost-effectiveness and energy cost-effectiveness refers to the reciprocal of power cost and energy cost, respectively). **b** Costs of major components and the total stack cost. **c** Costs of anolyte and catholyte and total electrolyte cost. Reference refers to data reported in ref. [24].

catholyte, the reactant is $OH^-$ rather than $H^+$ in acidic catholyte. However, all available AEMs are vulnerable to polysulfide crossover[28]. A solution to this is to use separate OER and ORR air electrodes for the charge and discharge process respectively, which was the solution of previous state-of-art work[24], which of course would add to the total cost and the complexity of the battery operation.

On the other hand, although it is applicable to use an acidic-catholyte-alkaline-anolyte PSA RFBs, as presented in previous state-of-art work[24], the neutralisation of two electrolytes with different pH prevents the use of any commercially available membrane. In previous work, a LISICON membrane was used, further adding to the total cost.

In this work, the dual-membrane design presented mitigates these issues. It allows the use of a single air electrode for both ORR and OER. It also allows the use of commercially available membranes to achieve both polysulfide crossover resistance and sufficient reactant for alkaline oxygen reactions, which no single membrane can satisfy. Ultimately, a stable alkaline PSA RFB is built round it, and has proven to lower the cost of this system.

The dual-membrane design has the potential to give further flexibility to the design of polysulfide-based RFB and other types of RFBs: Species crossover has long been a challenging problem for all types of RFBs[17,46,47]. And different requirements for membranes apply for different types of RFB. When it is difficult or too costly to find a single competent membrane, dual-membrane structures with appropriate combinations of membranes could offer an effective solution.

Furthermore, the intermediate solution is functioning mainly as a source/sink of $OH^-$ in this work, it can also be used as a reactor that contains appropriate chemical or physical processes to improve battery performance in future battery designs. For example, in polysulfide-based RFBs, a replaceable $MnO_2$ filter can be added to the cycling loop of the intermediate solution, which can adsorb any polysulfide ion that crossover from the anolyte side before it crosses to the catholyte[48], further mitigating the polysulfide crossover; Similarly, additives can be added to the intermediate solution to react with any crossed-over species to "anchor" them by forming complexes or precipitation, for example PVP-iodine complex[49], polysulfide-metal complex[50] etc.; Other suitable additives can also be added to the intermediate solution to enhance stability and performance of RFB batteries[51] without affecting the electrolyte chemistry.

To summarise, the dual-membrane structure adds more flexibility and options in the design of polysulfide-based and other RFBs. Future work will focus on exploring the potential of this structure for use in other flow battery concepts.

In summary, we have demonstrated an all-alkaline polysulfide-air redox flow battery (PSA RFB) system, employing aqueous

PSOR/PSRR and alkaline-based OER/ORR as the negative and positive redox couples, which is predicted to have an exceptionally low energy cost (~2.54 US$/kWh). The dual-membrane cell design, employing an AEM and a CEM in a single cell, proved effective in mitigating the crossover of (poly)sulphide species towards the air-side half-cell. The alkaline PSA RFB proposed in this work achieved a maximum power density of 5.81 mW cm$^{-2}$ at 55 °C, using commercially available electrodes and membranes, which is higher than the previous state-of-the-art PSA RFB system using precious metal catalyst and solid-state electrolyte separator[24]. Charge and discharge cycling at current densities up to 5 mA cm$^{-2}$ were performed on this alkaline PSA RFB. An average round-trip energy efficiency of 40% was achieved at 1 mA cm$^{-2}$ over 80 cycles. The energy cost and power cost of this PSA RFB are 26% and 20% of the previous state-of-art PSA system. The efficiency loss was mainly attributed to the rising OER overpotential of the $MnO_2$-based air electrode. Hence, future studies should focus on developing better electrodes, especially a highly active and reversible air electrode. The dual-membrane structure has proved to be an effective design to mitigate polysulfide crossover and lower overall cost. Further work will be required to develop more selective IEMs towards the polysulfide species as well as improving chemical stability in alkaline electrolytes[17,28,52], and on exploiting the potential of dual-membrane structure for different RFB systems. This work will also inspire the design and continued optimisation of the PSA RFB system to meet longer-term grid-scale energy storage demand for renewable energy development.

## Methods

**Preparation of electrolytes.** Sodium hydroxide (NaOH, ≥98%), sodium sulphide nonahydrate (Na$_2$S·9H$_2$O, ≥98%) and sulfur (S, ≥99.5%) were purchased from Sigma-Aldrich and used as-received. 1 M NaOH solution was prepared by directly dissolving a quantity of NaOH in deionized water (Millipore, 10 MΩ). 1 M Na$_2$S$_2$ + 1 M NaOH solution was prepared by dissolving a stoichiometric ratio of Na$_2$S·9H$_2$O and S into 1 M NaOH solution. 0.1 M Na$_2$S$_2$ + 1 M NaOH solution was prepared by diluting the 1 M Na$_2$S$_2$ + 1 M NaOH NaOH solution with 1 M NaOH solution.

**Pre-treatment of Ni foam.** Ni foam (purity 99.5%, porosity 95%, thickness 1.6 mm, bulk density 0.45 g cm$^{-3}$, average pore size 500 μm) was purchased from Goodfellow and cut into 1 × 3 cm$^2$ and 2.23 × 2.23 cm$^2$ sizes for the three-electrode cell and RFB measurements, respectively. Pre-treatment of the Ni foam included first sonicating in 6 M hydrochloric acid (HCl) for 30 min and then boiling in 1 M Na$_2$S$_2$ + 1 M NaOH solution for 30 min. The sulfidised Ni foam was stored in fresh 1 M Na$_2$S$_2$ + 1 M NaOH solution before characterisation.

**Pre-treatment of membranes.** Nafion 117 CEM (178 μm) and Fumasep FAA-3-PK-130 AEM (130 μm) were purchased from Fuel Cell Store. Nafion 117 CEM was pre-treated by soaking firstly in a 5 wt.% H$_2$O$_2$ solution at 80 °C for 30 min and then in 1 M NaOH solution at ambient temperature for at least 24 h, in order to convert it from the H$^+$-type (i.e., proton conductive) to the Na$^+$-type (i.e., sodium

ion conductive). The Fumasep FAA-3-PK-130 AEM was pre-treated by allowing it to pre-expand in a 1 M NaOH solution followed by heating it in a separate 1 M NaOH solution at 80 °C for 2 h, to convert it from the Br⁻-type (i.e., bromide conductive) into the OH⁻-type (i.e., hydroxide conductive). Both pre-treated Nafion 117 CEM and Fumasep FAA-3-PK-130 AEM were stored in 1 M NaOH solution before cell assembly.

**Characterisation of Ni foam electrode.** The morphology of the as-received and pre-treated Ni foams were characterised by scanning electron microscopy (SEM, LEO Gemini 1525 FEGSEM) with an in-lens detector and 5 kV acceleration voltage. The crystallinity and crystal size of the nickel foams were characterised by X-ray powder diffraction (XRD, X'Pert³ Powder, Malvern Panalytical). The surface substance composition of the pre-treated Ni foam was investigated by X-ray photoelectron spectroscopy (XPS, K-Alpha+, Thermo Scientific), operating at $2 \times 10^{-9}$ mbar base pressure, 6 mA emission current and 12 kV anode bias, with a flood gun to minimise the sample charging. Data were collected at 20 eV pass energy for core levels and 200 eV for the survey spectra using an X-ray spot size of 400 μm.

An in-house three-electrode cell was constructed to perform the polarisation measurements of the as-received and pre-treated Ni foams in the 1 M $Na_2S_2$ + 1 M NaOH solution, as illustrated in Supplementary Fig. 1, comprising a 100 ml three-neck flask (VWR), a working electrode holder with Pt current collector, an Ag|AgCl reference electrode with salt bridge (Metrohm) and filled with 3 M KCl solution saturated with AgCl (Sigma-Aldrich), and a Pt counter electrode. N₂ was purged into the electrolyte for 5 min after assembling the three-electrode cell to remove all dissolved oxygen within the aqueous polysulfide electrolyte followed by an N₂ blanket above the electrolyte surface throughout the measurements to prevent atmospheric oxygen from entering the three-electrode cell.

**Diffusivity of aqueous polysulfide ions through ion-exchange-membranes by UV-Vis absorption spectroscopy.** The flux of aqueous polysulfide ions through the single-membrane structure with either Nafion 117 CEM or FAA-3-PK-130 AEM were investigated using an H-cell setup. The diffusivities of aqueous polysulfide ions through the dual-membrane structure with both CEM and AEM were investigated in an actual RFB, as shown in Fig. 3a. In both experiments, the set-ups were kept in an argon-filled glovebox during the measurements. Before measurement, a calibration curve was obtained by measuring the absorbance of a number of dilute polysulfide solutions, with a concentration range of 0.5−10 mM $Na_2S_2$. All solutions contained 1 M NaOH supporting electrolyte in order to exclude the effect of electric field on the diffusivity of (poly)sulfide ions.

In the crossover measurement of single-membrane structure, 50 mL 0.1 M $Na_2S_2$ + 1 M NaOH solution and 50 mL 1 M NaOH solution were used as the feed and permeate solutions, respectively. The active area of the membranes within the H-cell setup was 1.54 cm². In the crossover measurement of the dual-membrane structure, three pumps were connected with the anode chamber, intermediate chamber and cathode chamber, respectively. In all, 50 mL 0.1 M $Na_2S_2$ + 1 M NaOH solution was circulating in the anode chamber, 50 mL 1 M NaOH in the intermediate chamber, argon gas was circulating in the cathode chamber to mimic the real battery operation condition. All pumps were operating at 50 mL min⁻¹ pumping rate. A UV-vis spectroscope (UV1800, Shimadzu) was used to determine the amount of permeated polysulfide ions. The error of measurement is ±0.7% according to the manufacturer's website information.

The cumulative flux $J_C$ of polysulfide species is determined by the following equation:

$$J_C = \frac{c}{A} \quad (4)$$

Where $c$ is polysulfide concentration, determined by UV-vis spectrum, $A$ is the crossover area.

The flux is determined by the change of cumulative flux through time:

$$J = \frac{J_c}{t} \quad (5)$$

**RFB setup.** The single-membrane-designed alkaline PSA RFB was assembled based on a commercially available 5 cm² RFB test fixture (Scribner Associates), as illustrated in Supplementary Fig. 6a. The dual-membrane-designed PSA RFB was constructed by adding an additional membrane, a polypropylene frame, and 13 layers of glass fibre filter paper (GF/A grade, thickness 0.26 mm, Whatman) as the flow channels for NaOH electrolyte into the single-membrane-designed setup, as illustrated in Supplementary Fig. 6b. Figure 4a, b shows the photo of experimental setup of a 5 cm² single-membrane and dual-membrane-structured alkaline PSA RFB, respectively. The sulfidised Ni foam and a commercial MnO₂-based air electrode (Gaskatel GmbH, 12 mg cm⁻² loading) were used as the anode and cathode, respectively. Pre-treated Nafion 117 and Fumasep FAA-3-PK-130 were mianly used as the CEM and AEM, respectively. Pre-treated Nafion 117 and PAP-TP-100[41] (also known as PiperION commercially) were also used to test the influence of different membranes. The aqueous electrolytes were circulated by peristatic pumps (Masterflex, Cole-Parmer) at a flow rate of 50 ml min⁻¹. CO₂-free

Air (BOC) was humidified by a gas bubbler and then supplied into the cathode compartment at a flow rate of 100 ml min⁻¹ controlled by a mass flow controller (F-201CV, Bronkhorst). The operating temperature of this PSA RFB was controlled by a fuel cell tester (855, Scribner Associates). In the dual-membrane setup, an Hg|HgO reference electrode (L120-S7, Origalys) filled with 1 M NaOH solution was placed into the NaOH electrolyte vessel to decouple the potentials of the negative and positive half-cells. An MX100 Data Acquisition Unit (Yokogawa) was used to monitor and record the potentials of the full-cell and both half-cells in parallel to the electrochemical measurements conducted by the potentiostat.

**Electrochemical characterisation.** Electrochemical measurements were performed using an Autolab potentiostat/galvanostat (PGSTAT302N, Metrohm). Polarisation measurements of the Ni foams in 1 M $Na_2S_2$ + 1 M NaOH electrolyte was conducted using the linear scan voltammetry (LSV) method in the overpotential window of −200–200 mV with a scan rate of 1 mV s⁻¹ as reported earlier[21]. Polarisation curve measurements of the PSA RFB were performed using the chronopotentiometry method, where the anolyte and catholyte used 10 mL 1 M $Na_2S_2$ + 1 M NaOH solution and 50 mL 1 M NaOH solution, respectively. For each polarisation measurement, a constant current was applied to the flow battery for 1 min where the steady-state potential at the end of the measurement was recorded, after which the battery was charged back to the original SOC under the same current density before proceeding to the next polarisation measurement at a higher current density. Potentiostatic EIS of the full cells was measured at OCV at 100 SOC. The input signal was sinusoidal with 20 mV amplitude, the oscillation frequencies ranged from 100,000 to 0.02 Hz. 50 data points were obtained in each tests.

Charge/discharge cycling of the flow battery was also performed using the chronopotentiometry method, where the same current density (1–5 mA cm⁻²) was applied to the charge and discharge processes in each cycle. All full battery cycling in this work was performed using 100% theoretical capacity of $S_2^{2-}/S_4^{2-}$ (26.8 Ah L⁻¹ for 1 M $Na_2S_2$ + 1 M NaOH and 2.68 Ah L⁻¹ for 0.1 M $Na_2S_2$ + 1 M NaOH) unless the battery reached the cutoff voltage before reaching the theoretical capacity. The anolyte and catholyte used in the cycling experiments were 5 mL 0.1 M $Na_2S_2$ + 1 M NaOH solution and 50 mL 1 M NaOH solution, respectively. Upon the use of FAA-3-PK-130 AEM, an upper cutoff voltage of 1.4 V was set to prevent significant degradation of the AEM. All measurements were performed at 25 °C, unless mentioned otherwise. In the testing of dual-membrane PSA RFB, 1 M NaOH solution was used as the intermediate solution, a Hg|HgO reference electrode (L120-S7, Origalys) filled with 1 M NaOH solution was inserted in the intermediate solution to provide the voltage profile for each half cell.

**Reporting summary.** Further information on research design is available in the Nature Research Reporting Summary linked to this article.

## Data availability

The data that support the findings of this study are available from the corresponding authors upon reasonable request.

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

## Acknowledgements

The authors gratefully acknowledge financial support from the EPSRC for projects EP/L014289/1 and EP/K002252/1. The authors would also like to thank RFC Power Ltd for the technical discussion. This project has received funding from the European Research Council (ERC) under the European Union's Horizon 2020 research and innovation programme (grant agreement no. 851272, ERC-StG-PE8-NanoMMES). R.T. acknowledges a full Ph.D. scholarship funded by the China Scholarship Council. A.R. acknowledges support from the Analytical Chemistry Trust Fund for her CAMS-UK Fellowship and from Imperial College London for her Imperial College Research Fellowship.

## Author contributions

Y.X., N.B., V.Y. and M.O. conceived the project and designed the experiments. Y.X., M.O., A.R., R.T, A.W. and B.C. performed the materials characterisations. Y.X. and A.K. designed and manufactured the flow battery cell components. Y.X., M.O. and W.M. conducted the flow battery tests. Q.S. helped with experimental design and edited the manuscript. Y.X. and M.O. performed the economic analysis. N.B., M.O. and V.Y. supervised the project. Y.X. and M.O. wrote the original manuscript. A.K. provided help with experiment design and development of the manuscript. M.O. led the revision of the manuscript. M.O., R.T. and A.W. performed additional experiments in revision. M.O. and Y.X. wrote the reply to the reviewers. All authors analysed the data and contributed to the review of the manuscript.

## Competing interests

N. Brandon, A. Kucernak and V. Yufit are authors on a patent (no. PCT EP2011/066238) related to this work held by RFC Power, a spin-out company from Imperial College London founded by N. Brandon, A. Kucernak and V. Yufit developing long-duration flow battery technologies including one related to this work. All other authors declare no competing interests.
