## [Peer Review File · Nature Communications]

REVIEWER COMMENTS

Reviewer #1 (Remarks to the Author):

This manuscript presents an alkaline polysulfide (PS)-air redox flow battery assembled with a cation exchange membrane (Nafion 117) and anion exchange membrane (Fumasep FAA). The authors demonstrated that the dual-membrane cell design can reduce PS crossover and improve OH⁻ transport, thus enhancing coulombic efficiency (CE) and voltage efficiency (VE). They also systematically characterized electrochemical properties of electrodes and membranes, and evaluated cell performance. Overall, the dual-membrane RFB cell showed better performance than a single-CEM and AEM RFB cell. However, to publish this manuscript, the following points should be considered by the authors.

- 1) Although the role of ion exchange membranes is critical in the dual-membrane cell RFB system, the authors evaluated the performance of the RFB cell assembled with only two commercial AEM and CEM rather than systematically investigating the effect of ion-exchange membranes. The authors used 180-um-thick Nafion 117 membrane over Nafion 115 or 212, which resulted in high CE but slightly low VE, as shown in Figures 4f and 4g. Why didn't the authors use thinner CEM (e.g., 130-um-thick Nafion 115)? Although the CE value of the cell with Nafion 117 could be similar or slightly lower than 100%, much higher conductance could increase the VE of the cell with Nafion 115.
- 2) Figure 4: why did the authors show only 20 cycles of the cell with CEM, although VE, CE, and EE values are slightly lower than the cell with AEM/CEM? Also, the authors showed only 30 cycles that is not the standard cycle stability test. To support the authors' claim about the stability of the alkaline PSA RFB, the authors should show cycle stability with a charge capacity retention curve longer than a minimum of 100 cycles
- 3) Page 6, line 150: In Figure 2p and Table S1, after the sulfidisation, the cathodic exchange current density is greatly increased compared to the anodic side. Although the authors briefly mentioned that the catalyst follows different mechanisms and citation, it is highly recommended to provide a more detailed explanation.
- 4) Page 9, line 205: the authors showed overall resistance of AEM and CEM using NaOH. What are the Na⁺/OH⁻ transport rate ratio (i.e., transference number, selectivity) of AEM and CEM?
- 5) Page 11, line 266: the authors say, "the dual-membrane cell structure was successful in mitigating the crossover of polysulfide ions towards the air side half-cell". However, the PS crossover rate is greatly dependent on not CEM/AEM but CEM. The authors should revise the sentence.
- 6) Page 21, line 569: the cycling test was performed with 0.0-1.4 V of cut-off voltage and cut-off capacity of the theoretical capacity of polysulfide anolyte. 7 mL of low concentration (0.1 M) of polysulfide anolyte was used to reduce the charge/discharge time with the theoretical capacity of 6.7 Ah/L. Thus, the theoretical capacity is around 46.9 mAh (= 6.7 Ah/L * 7 mL). In the charge/discharge curve figures in this manuscript, the battery capacity was cut at around 13.5 mAh. How was this value calculated?
- 7) The theoretical volumetric capacity of the RFB cell with 0.1M of polysulfide concentration is 6.7 Ah/L, which is lower than 15 – 28 Ah/L of the typical vanadium redox flow battery (VRFB). To compete against VRFB, a higher concentration of PS in anolyte is more reasonable (e.g., > 1M for 67 Ah/L). However, it is expected that osmotic water flows in the cell if 1 M Na₂S₂/1 M NaOH and 1M NaOH are used, which can affect the transport of ions and could cause the cell failure. The authors should address this issue.
- 8) As mentioned in line 74, the kinetic of dissolved polysulfides in an aqueous solution is complex with varieties of sulfur species and ions. The bisulfide ion (HS⁻) could further hydrolyze and produce H₂S as the anolyte pH changes during the charge and discharge, which can cause the loss of active sulfur material as the gaseous product. Did the authors observe this issue during the experiments?

Minor: the authors should carefully check the figure captions and figure numbers in the main text. For example,

- 1) Page 6, line 161: "f,g,h SEM images of the as-received Ni foam. h, SEM image of the acid-treated Ni foam. i, photo of the sulfidised Ni foam." -> "f,g, SEM images of the as-received Ni foam. h, SEM image of the acid-treated Ni foam. i, photo of the sulfidised Ni foam."**
- 2) Page 9, line 206: Figure.3 (a) should be Figure.3 (f)**
- 3) Page 9, line 209: Figure.3 (c) should be Figure.3 (f)**

Reviewer #2 (Remarks to the Author):

The authors used a dual-membrane (CEM+AEM) cell architecture to design a low-cost alkaline polysulfide-air redox flow battery. But, the necessity of AEM in this work is insufficient based on the authors' discussion. CEM is a very common strategy to reduce the shuttle of polysulfides, which has been used in many works for designing polysulfide-based redox flow batteries, so, the advantages of AEM in here should be more described and verified rather than CEM. It is not recommended for publication on Nature Communications in the current status.

- 1. Is AEM necessary in Fig. 1c? As a very common strategy to reduce the shuttle of polysulfides, CEM has been used in many works for designing polysulfide-based redox flow batteries. Therefore, the advantages of CEM in here does not need to be described too much. Unlike other works, the authors used dual-membrane (CEM+AEM) architecture, so, more description about the importance of AEM should be emphasized in the article. If AEM is not added, can the hydroxide conductivity be increased by directly turning the positive side into a flowing state? If so, why adding AEM?**
- 2. The comparison between Fig. 1b and Fig. 1c is incompletely correct. In Fig. 1c, the flow electrolyte on the positive side will increase hydroxide diffusion. However, the positive side in Fig. 1b is static. So, the comparison of hydroxide diffusion between Fig. 1b and Fig. 1c is incorrect. How to determine whether it is caused by adding AEM rather than flow? Therefore, the positive side in Fig. 1b should be operated under the flow condition without AEM. This can highlight the importance of AEM.**
- 3. The Ref. 25 also used sodium salts (Na₂S₄, Na₂SO₄ and NaOH). So, the authors should also indicate sodium salts, not just lithium salts, when describing this work.**
- 4. For Fig. 3, the authors compare the barrier performance of CEM and AEM on polysulfides. However, it can only explain that the cell structure in Fig. 1b is better than that in Fig. 1a, and cannot explain the advantages of adopting the cell structure in Fig. 1c.**
- 5. In Fig. 4, as described above, when only CEM is used, the positive side is static, and the double membrane is used, the positive side is flow. The flow electrolyte has a great influence on the hydroxide diffusion, so, the results in Fig.4 cannot reflect the advantages of adding AEM. Why not use the same flow mode to evaluate the necessity of AEM? So, the comparison of electrochemical performance should be carried out with/without AEM using the same flow mode.**
- 6. In Fig. 6, the comparison of costs with Ref. 25 is hard to be persuasive. Two types of salts (Na/Li) are used in Ref. 25. If the price of Na salts is comparable with that of Li salts, it is obvious that the price of Na salts is cheaper than that of Li salts, and it has been reflected in Ref. 25 (Fig. 1). However, in Fig. 6b, the authors claim that the catholyte used in this work is cheaper than Ref. 25. Why? If it is all Na salts, what is the price difference between the two? The types of Na salts are different?**

Reviewer #3 (Remarks to the Author):

This manuscript reports an experimental study of a dual-membrane PSA RFB, where a pair of AEM and CEM are installed. The device's configuration is effective and straightforward, but the implementation is incremental. Cross-over is significantly reduced, which makes the use of inexpensive catalysts possible. Relatively higher performance (than similar but single membrane setup) is obtained, although overall the power/energy densities, as well as lifetime (performance decay), are still much lower than incumbent technologies. The manuscript is well structured and clearly written. The design of this dual-membrane shows great potential for future development and application. Some comments about this manuscript are as follows.

- 1. Keep a reasonable number of figures and use only necessary images, e.g., Fig. 2 and Fig. 3. Some images are subjective, e.g., color change in Fig. 3. Also, cartoons in Fig. 3 are redundant and can be described in the text.**
- 2. It is felt that a cost analysis is somewhat premature for this topic since the performance is rather low and the decay rate is high. Efforts should be directed to gain an understanding of the mechanisms of degradation and possible approaches to enhance such design's performance and lifetime, as authors mentioned in lines 313-315 and lines 355-357.**
- 3. Fig. 4e and Fig. 5e show voltage variation vs. capacity for different cycles. It is curious that the results of the 10th cycle and 20th cycle appear to be very close to each other. Please provide an explanation and discussion.**
- 3. The tested current densities in this study are low, thus diffusion of species appears to be the predominating mechanism of transport. Authors might also want to consider other mechanisms such as electro-osmotic drag, in particular on how these transport mechanisms may affect the (limiting) air-side electrode. Please elaborate.**

We are grateful to all reviewers for your valuable and constructive comments on our manuscript. We have addressed all of those comments and performed additional experiments and analyses. A brief summary of revision is shown in the List of corrections.

We believe that the quality of the revised manuscript is improved and suitable for publication.

Below we list the detailed comments (black) and our point-to-point response (blue).

REVIEWER COMMENTS

Reviewer #1 (Remarks to the Author):

This manuscript presents an alkaline polysulfide (PS)-air redox flow battery assembled with a cation exchange membrane (Nafion 117) and anion exchange membrane (Fumasep FAA). The authors demonstrated that the dual-membrane cell design can reduce PS crossover and improve OH⁻ transport, thus enhancing coulombic efficiency (CE) and voltage efficiency (VE). They also systematically characterized electrochemical properties of electrodes and membranes, and evaluated cell performance. Overall, the dual-membrane RFB cell showed better performance than a single-CEM and AEM RFB cell. However, to publish this manuscript, the following points should be considered by the authors.

1) Although the role of ion exchange membranes is critical in the dual-membrane cell RFB system, the authors evaluated the performance of the RFB cell assembled with only two commercial AEM and CEM rather than systematically investigating the effect of ion-exchange membranes. The authors used 180-um-thick Nafion 117 membrane over Nafion 115 or 212, which resulted in high CE but slightly low VE, as shown in Figures 4f and 4g. Why didn't the authors use thinner CEM (e.g., 130-um-thick Nafion 115)? Although the CE value of the cell with Nafion 117 could be similar or slightly lower than 100%, much higher conductance could increase the VE of the cell with Nafion 115.

Response: Thank you for your suggestions. We used the thick Nafion 117 instead of Nafion 115 and Nafion 212 to minimise the polysulfide crossover¹. As we have explained in page 13 in the manuscript, the air electrode reaction is the rate-limiting part of the whole battery operation. This is proved by the EIS of the cell shown in Figure S9, in which the Ohmic resistance indicated by the intercept on the Z_{real} axis is $1.5 \Omega \text{ cm}^2$, and is 1/30 of the charge transfer resistance. It suggests that the resistance due to ion transport through the membrane only contributes a small part of the whole cell resistance, and that switching to a thinner membrane would not significantly lower the polarisation of the whole cell, at least until better electrodes have been developed. We agree that additional tests with thinner CEM membranes can demonstrate this point more clearly.

Similarly, switching to different commercial membranes should have little effect on the current performance of the cell. We chose Nafion and FAA-3 because these are the most commonly used CEM and AEM available on the market, and can show that the dual membrane structure is universal and does not require the use of particular membrane. We agree with reviewer's suggestion that testing more commercial membrane can demonstrate the generic approach.

Therefore, to address the above two points, we tested two dual membrane structure cells, the first using Nafion 212 as CEM and FAA-3 as AEM, the second using Nafion 117 and a newly developed AEM PAP-TP-100² (commercially available as PiperION). The EIS and first 10 cycles of cells using different combinations of membranes are shown in **figure S12**. The Ohmic resistance of Nafion 212/FAA-3-PK-130 is 4% lower than that of Nafion 117/FAA-3-PK-130 ($2.55 \Omega \text{ cm}^2$), while the Nafion 117/PAP-TP-100 is 5% higher. These differences can be neglected compared with the total

impedance which is over $60 \Omega \text{ cm}^2$. The first 10 cycles of different combinations were also similar. This confirmed our finding that the rate-limiting part of this system is the electrode, particularly the air electrode. These additional results are described in pages 15-16 of the revised manuscript:

“Nafion 117 and FAA-3 were chosen in this work because these are some of the cheapest and most commonly used CEMs and AEMs in the energy storage field. However, dual membrane cells using two other commercially available membranes were also assembled and tested; the first using Nafion 212 as CEM and FAA-3 as AEM, the second using Nafion 117 and a newly developed AEM PAP-TP-100 (commercially available as PiperION). The EIS and first 10 cycles of cells using different combinations of membranes are shown in Figure S12. The Ohmic resistance of Nafion 212/FAA-3-PK-130 was 4% lower than Nafion 117/FAA-3-PK-130 ($2.55 \Omega \text{ cm}^2$), while Nafion 117/PAP-TP-100 was 5% higher. These differences are small compared with the total impedance which is over $60 \Omega \text{ cm}^2$. The polarisation and degradation of the first 10 cycles of different membrane combinations are also similar. This suggests that changing the membranes had little influence on the performance of the dual membrane PSA RFB. This supports the view that the concept is not restricted to the use of particular membranes, and further validates our conclusion that the current rate-limiting part of this system is the electrode, particularly the air electrode.”

2) Figure 4: why did the authors show only 20 cycles of the cell with CEM, although VE, CE, and EE values are slightly lower than the cell with AEM/CEM? Also, the authors showed only 30 cycles that is not the standard cycle stability test. To support the authors' claim about the stability of the alkaline PSA RFB, the authors should show cycle stability with a charge capacity retention curve longer than a minimum of 100 cycles

The charging plateau of the CEM cell reaches $\sim 1.8 \text{ V}$ eventually, which would cause dissolution of the MnO_2 cathode³. As shown in **figure 4d** the average charging plateau reached 1.5 V after 20th cycle. This was the main reason we only did 20 cycles of the CEM cell. We did find the MnO_2 cathode was partially dissolved after cycling. When we flushed the cathode with deionised water, the water turned purple, suggesting the existence of dissolved Mn. To make the reason clearer, we have added a discussion on page 12 of the manuscript.

We agree with the reviewer that longer cycling is needed to demonstrate the stability of the dual-membrane cell. Therefore, we repeated the cycling experiment up to 80 cycles across 429 h.

The degradation was mainly observed on the charging process after 80 cycles. The charging curve reached the cutoff voltage of 1.4 V and the capacity started to fade. The processed data has replaced **figure 4e, f, g, h** and **figure 5d, e** in the revised manuscript, **figure S11, table S2** and **table S7** in the revised SI. Originally, we changed the polysulfide solution every five cycles to eliminate the effect of polysulfide oxidation by leaked air to the degradation of the cell. In the new cycle experiment we changed the solution every ten cycles instead of five. The detailed discussion is shown on page 15 of the revised manuscript. Half-cell voltage profile in **figure 5 d,e** shows that the rising charging plateau is mainly attributed to the increasing OER polarisation on cathode side. We think the main cause of degradation is the gradual dissolution of MnO_2 .

The revised figures in the manuscript are shown below:

Figure 4. Charge and discharge cycling of the alkaline PSA RFB. a, experimental setup of single-membrane-structured PSA RFB. b, experimental setup of dual-membrane-structured PSA RFB. c, voltage profile of the PSA RFB with FAA-3-PK-130 AEM over 10 cycles. d, voltage profile of the PSA RFB with Nafion 117 CEM over 20 cycles. e, voltage profile of the PSA RFB with both FAA-3-PK-130 AEM and Nafion 117 CEM over 80 cycles. f, voltage efficiencies against cycle number of the PSA RFBs with either single or dual membranes. g, coulombic efficiencies against cycle number of the PSA RFBs with either single or dual membranes. h, round-trip energy efficiencies against cycle number of the PSA RFBs with either single or dual membranes.

The discussion added on page 15 is shown below:

Figure 5. Polarisation curve measurement and cycling performance of the alkaline PSA RFB with dual membranes. a, polarisation curve and power density curve at 25 °C and various SOC. b, polarisation curve and power density curve at 50% SOC and various temperatures. c, measured half-cell overpotentials against current density at 25 °C and 50% SOC. d, measured polysulfide-side half-cell potential profile against capacity. e, measured air-side half-cell potential profile against capacity. f, round-trip energy efficiency against current density.

“The upper or the lower cut-off voltages were not reached during battery cycling until the 80th cycle, as shown in Figure 4 (e). Hence, the capacity and the coulombic efficiency of this alkaline PSA RFB were both maintained at 100% throughout the 79 cycles, and were 98% for the last cycle. Therefore, the round-trip energy efficiency of this alkaline PSA RFB was calculated to be the multiplication of its voltage efficiency and coulombic efficiency. Figure 4 (h) presents the round-trip energy efficiencies of this alkaline PSA RFB as a function of the cycle number. The repeating pattern on the efficiency figure is attributed to replenishing the polysulfide anolyte every 10 cycles to eliminate the effect of polysulfide oxidation by leaked air.”

3) Page 6, line 150: In Figure 2p and Table S1, after the sulfidisation, the cathodic exchange current density is greatly increased compared to the anodic side. Although the authors briefly mentioned that the catalyst follows different mechanisms and citation, it is highly recommended to provide a more detailed explanation.

This is a very interesting topic. The sulfidised nickel has been a standard catalyst in polysulfide-based system due to the great enhancement of polysulfide cathodic reaction kinetics^{4,5}. There has been some detailed analysis of the polysulfide reaction mechanism on sulfidised metal in literatures^{6,7}. Since this work is mainly on the benefit of dual-membrane structure, we would like to focus on this topic. The mechanism study of polysulfide catalyst is indeed very important, but is out of the scope of this work, and we will aim to study it in the future.

We have added some discussion on page 6-7 of the revised manuscript based on our observations and references to literature work:

“The overall performance enhancement of sulfidised nickel can be explained by the enhanced electron transfer from the strongly coupled Ni/NiS_x interface and the much faster polysulfide adsorption/desorption rate from the transient metal sulfide surface compared with the partly sulfidised metal surface. The more significant improvement on the cathodic branch is consistent with previous observations. This indicates NiS_x plays a vital role in the electrochemical reduction of polysulfide species. However, detailed analysis of this phenomena merits in-depth study in the future.”

4) Page 9, line 205: the authors showed overall resistance of AEM and CEM using NaOH. What are the Na⁺/OH⁻ transport rate ratio (i.e., transference number, selectivity) of AEM and CEM?

According to literature, the Na⁺ transference numbers of Nafion 117 in different concentration of NaCl solution range from 0.88 to 0.95^{8,9}. The Na⁺ transference numbers of Nafion 117 in 1 M NaOH is ~0.9¹⁰.

The OH⁻ transference number of FAA-3-PK-130 cannot be found in the literature. In the technical data sheet on the manufacturer’s webpage, FAA-3-PK-130’s Cl⁻ selectivity in 0.5 M KCl aqueous solution is 0.93 - 0.98, and hence we expect the transference number for OH⁻ to be similar if not even higher.

5) Page 11, line 266: the authors say, “the dual-membrane cell structure was successful in mitigating the crossover of polysulfide ions towards the air side half-cell”. However, the PS crossover rate is greatly dependent on not CEM/AEM but CEM. The authors should revise the sentence.

We have measured the polysulfide crossover through single AEM and CEM cells, as shown in figure 3 in the manuscript. To investigate the effectiveness of the dual-membrane structure in mitigating crossover, we performed additional experiment to measure the polysulfide crossover in a dual-membrane cell. The experiment is described in the method part of the revised manuscript on page 21 and the result of the experiment is shown and discussed in detail in **figure 3** and page 8-10 of the revised manuscript. **Figure 3** shows that polysulfide flux is lower across dual membrane structure than single CEM, suggesting the addition of AEM is useful to block crossover of low concentration polysulfide – this is probably because it acts as a diffusion barrier and might also be associated with some degree of size selection of anion.

The revised figure and discussion are shown below, **figure 3** is also modified according to reviewer 3’s suggestion:

Figure 3. Diffusion of polysulfide species through ion-exchange membranes. a, H-cell setup for polysulfide crossover determination on single AEM and CEM; redox flow battery setup for setup for polysulfide crossover determination on dual membrane structure. b, Area specific polysulfide concentration of the permeate solution in single-AEM, single-CEM and dual-membrane setup against permeation time. c, Calculated polysulfide permeance across single-AEM, single-CEM and dual-membrane structure. d, EIS of CEM and AEM. e, the polysulfide permeance versus membrane conductivity.

“The cumulative flux is the total amount of species transported across the membrane normalised by the effective membrane permeation area. The straight line fitted against permeation time indicates a constant flux of polysulfide ions through the IEMs. The calculated flux of polysulfide ions through Nafion 117 CEM, FAA-3-PK-130 AEM are 3.50×10^{-4} and 1.15×10^{-2} cm² s⁻¹, respectively,

indicating a 32.9 times lower transportation rate of polysulfide ions through Nafion 117 CEM than FAA-3-PK-130 AEM. This is also demonstrated by the evidently slower colour deepening of the permeate solution over time in the H-cell setup assembled with Nafion 117 CEM. To conclude, FAA-3-PK-130 AEM would allow OH⁻ to pass through but is more vulnerable towards polysulfide crossover, while Nafion 117 CEM would allow Na⁺ to pass and have a higher resistance of polysulfide crossover but it would block OH⁻.

“To test the polysulfide crossover through the dual membrane structure, a dual-membrane redox flow battery was assembled in the glovebox, as shown in Figure 3(a). Three peristaltic pumps were used to circulate 50 mL 0.1M Na₂S₂/1M NaOH solution to the anode side, 50mL 1M NaOH solution to the intermediate plate, and environmental argon to the cathode side respectively to mimic the real PSA RFB operating conditions. All pumping speed were set to 50 mL min⁻¹. Every 25h, polysulfide concentration in the intermediate solution was examined using UV-vis analysis, denoted as Dual-mid. At the same time, the cathode side was flushed by 50 mL of 1M NaOH for 2 min, and the polysulfide crossover amount on the cathode side was determined by examining the polysulfide concentration of this flushed solution, denoted as Dual-air. The increase of total amount of polysulfide in the permeate solution normalised by area with time is shown in Figure 3(b), and the polysulfide flux through the dual-membrane structure is shown in Figure 3(c). The amount of polysulfide crossover in the intermediate solution of the dual membrane structure is identical with that in permeate solution of the single-CEM structure, suggesting the crossover experiments using different setup (H-cell vs redox flow battery) are comparable and both are valid. On the other hand, the polysulfide crossover rate through the dual-membrane structure is much lower than that of single-CEM; the calculated flux (Figure 3(c)) is 4% of the CEM and 0.1% of the AEM. This suggests that the dual-membrane structure is effective in blocking polysulfide crossover, and that the AEM can be used to block small concentrations of polysulfide ions.”

6) Page 21, line 569: the cycling test was performed with 0.0-1.4 V of cut-off voltage and cut-off capacity of the theoretical capacity of polysulfide anolyte. 7 mL of low concentration (0.1 M) of polysulfide anolyte was used to reduce the charge/discharge time with the theoretical capacity of 6.7 Ah/L. Thus, the theoretical capacity is around 46.9 mAh (= 6.7 Ah/L * 7 mL). In the charge/discharge curve figures in this manuscript, the battery capacity was cut at around 13.5 mAh. How was this value calculated?

Thanks for pointing out the mistakes. The theoretical capacity of the 1M polysulfide anolyte should be 26.8 Ah/L, and 0.1M polysulfide anolyte should be 2.68 Ah/L, calculated as followed:

$$E = \frac{m_e - F \times C}{3600}$$

m_e - stands for mole of electron transferred per mole of S₂²⁻ reacted, which is 1 in this case; C is the concentration of S₂²⁻, mol/L; F stands for Faraday constant, As/mol.

The volume of 0.1M polysulfide we used to run the cycle should be 5 mL instead of 7 mL, which corresponds to 13.4 mAh total capacity. These has been corrected in the method section of the revised manuscript on page 22.

7) The theoretical volumetric capacity of the RFB cell with 0.1M of polysulfide concentration is 6.7 Ah/L, which is lower than 15 – 28 Ah/L of the typical vanadium redox flow battery (VRFB). To compete against VRFB, a higher concentration of PS in anolyte is more reasonable (e.g., > 1M for 67 Ah/L). However, it is expected that osmotic water flows in the cell if 1 M Na₂S₂/1 M NaOH and 1M NaOH are used, which can affect the transport of ions and could cause the cell failure. The authors should address this issue.

To demonstrate the feasibility of the polysulfide air flow battery, we used relatively low concentration of polysulfide at 0.1M, which indeed corresponds to relatively low volumetric capacity. However, as air is used as one of the reactants, the cost of that electrolytic species is effectively zero, and the

volume taken up by that species is also effectively zero. Hence when comparing to the vanadium system we save both the cost and “space” of one of the vanadium electrolytes. Furthermore, the low cost of sulphur compared to vanadium also mitigates against some of the limitations of this system. However, we agree with the referee that in a practical system it would be beneficial to operate with higher electrolyte concentration. In this case, the osmotic pressure of a high concentration polysulfide will very likely lead to water migration to the anolyte side and dilute the polysulfide solution, decreasing the OCV and thus the energy density of the whole battery¹¹. In fact, this osmotic pressure difference is well known in the forward osmosis and electro dialysis field. One literature work in vanadium flow batteries has proposed to balance the osmotic pressure in vanadium cell by adding soluble draw solute such as 2-methylimidazole¹². Similarly, we expect that the osmotic pressure in our dual membrane cell can be balanced by addition of electrochemically-inert salt or molecules as draw solute into the middle buffer solution. A discussion is added to the page 10-11 of the revised SI.:

“A potential problem associated with the future application of dual-membrane structures in PSA-RFBs is the osmotic effect, as identified in other RFB system. When the battery is cycled with high concentrations of polysulfide anolyte, the large concentration difference between the anolyte and the intermediate solution would result in an osmotic water migration from the intermediate solution to the anolyte, diluting the anolyte and impeding reverse hydroxide transportation. One literature work in vanadium flow battery has proposed to balance the osmotic pressure in vanadium cells by adding a soluble draw solute such as 2-methylimidazole. Similarly, we expect that the osmotic pressure in our dual membrane cell can be balanced by the addition of electrochemically-inert salt or molecules as draw solute into the middle buffer solution.”

The discussion is referred to on page 16 of the revised manuscript:

“Another potential problem associated with the future application of dual-membrane structures in PSA-RFBs is the osmotic effect, as identified in other RFB system. This is discussed in detail in supporting information III(i).”

8) As mentioned in line 74, the kinetic of dissolved polysulfides in an aqueous solution is complex with varieties of sulfur species and ions. The bisulfide ion (HS⁻) could further hydrolyze and produce H₂S as the anolyte pH changes during the charge and discharge, which can cause the loss of active sulfur material as the gaseous product. Did the authors observe this issue during the experiments?

We tested the pH of anolyte at 100% SOC and 0% SOC using a pH meter, noted in the table below.

	100 SOC	0 SOC
0.1 M Na ₂ S ₂ /1 M NaOH	15.1	15.3
1 M Na ₂ S ₂ /1 M NaOH	15.7	15.9

The pH of the anolyte is increasing during discharging process because of the water consumed and OH⁻ formed, $2S_2^{2-} + H_2O + O_2 \rightarrow S_4^{2-} + OH^-$. Higher pH will further suppress the formation of H₂S in the anolyte.

Even at 100% SOC the pH values of both concentrated and diluted anolytes are over 15, and the hydrolyzation of HS⁻ is greatly suppressed. According to the reference¹³, the equilibrium concentrations of H₂S are 3.27×10^{-8} M and 1.36×10^{-9} M in 1 M Na₂S₂/1M NaOH and 0.02 M Na₂S₂/1M NaOH respectively.

Also, during the experiment we didn't smell any H₂S, and the smell of H₂S is easily detectable at over 0.025 ppm¹⁴.

Therefore we are confident that the loss of sulfur active materials due to H₂S formation is extremely low.

Minor: the authors should carefully check the figure captions and figure numbers in the main text. For example,

1) Page 6, line 161: “f,g,h SEM images of the as-received Ni foam. h, SEM image of the acid-treated Ni foam. i, photo of the sulfidised Ni foam.” -> “f,g, SEM images of the as-received Ni foam. h, SEM image of the acid-treated Ni foam. i, photo of the sulfidised Ni foam.”

2) Page 9, line 206: Figure.3 (a) should be Figure.3 (f)

3) Page 9, line 209: Figure.3 (c) should be Figure.3 (f)

Thank you pointing these mistakes out. We have fixed the figure captions and numbers.

Reviewer #2 (Remarks to the Author):

The authors used a dual-membrane (CEM+AEM) cell architecture to design a low-cost alkaline polysulfide-air redox flow battery. But, the necessity of AEM in this work is insufficient based on the authors' discussion. CEM is a very common strategy to reduce the shuttle of polysulfides, which has been used in many works for designing polysulfide-based redox flow batteries, so, the advantages of AEM in here should be more described and verified rather than CEM. It is not recommended for publication on Nature Communications in the current status.

1. Is AEM necessary in Fig. 1c? As a very common strategy to reduce the shuttle of polysulfides, CEM has been used in many works for designing polysulfide-based redox flow batteries. Therefore, the advantages of CEM in here does not need to be described too much. Unlike other works, the authors used dual-membrane (CEM+AEM) architecture, so, more description about the importance of AEM should be emphasized in the article. If AEM is not added, can the hydroxide conductivity be increased by directly turning the positive side into a flowing state? If so, why adding AEM?

We agree with the reviewer that more experiments and discussion is needed to demonstrate the function of the AEM. Indeed the CEM has been applied to mitigate the crossover of anionic redox species. In the case of polysulfide species, there is still significant crossover when a single CEM is applied, as can be seen from previous literature¹⁵ and our polysulfide crossover experiment in **figure 3b**. To make better comparison and to investigate the effectiveness of the dual-membrane structure in mitigating crossover, we add a polysulfide crossover experiment using a dual-membrane cell. The experiment is described in the method part on page 21 and the result of the experiment is shown and discussed in detail in **figure 3** and page 9-10 of the revised manuscript. **Figure 3** shows that polysulfide permeance is lower across dual membrane structure than single CEM, confirming that the addition of AEM is useful to block crossover of low concentration polysulfide.

We also assembled the single-CEM battery operating in mixed-flow configuration. Apart from the membrane structure, all other parameters are the same as operating the dual-membrane cell. The pump used to circulate intermediate solution in the dual membrane cell is used to circulate 1 M NaOH solution through the cathode chamber, at the same time air is bubbled into the chamber through a three-way connector. The EIS and one cycle were tested on the flow state cell and compared with the dual-membrane cell. As shown in the figure below, the Ohmic resistance of the mixed flow single CEM cell is ~25% lower than dual-membrane cell (inset of the figure left), however the polarisation resistance of mixed flow cell is ~four times higher than dual membrane cell. Also shown is the charge-discharge curve: although the charge process shows less polarisation, the discharge process of a single CEM cell has a polarisation much higher compared to that of the dual-membrane cell. We think this is due to the liquid flow on the cathode side blocking the transport of oxygen to the catalytic active sites in the cathode. This is reasonable because the OER and ORR reactions on the oxygen cathode is determined by the local environment and gas/ion/electron transport. We will elaborate this in the next question.

Figure. Left: EIS of flow-state single CEM cell comparing with dual membrane cell (inset: magnified EIS plot); Right first cycle of flow-state single CEM cell comparing with dual membrane cell

2. The comparison between Fig. 1b and Fig. 1c is incompletely correct. In Fig. 1c, the flow electrolyte on the positive side will increase hydroxide diffusion. However, the positive side in Fig. 1b is static. So, the comparison of hydroxide diffusion between Fig. 1b and Fig. 1c is incorrect. How to determine whether it is caused by adding AEM rather than flow? Therefore, the positive side in Fig. 1b should be operated under the flow condition without AEM. This can highlight the importance of AEM.

In the single-membrane cells the membrane is directly in contact with the cathode, there is actually no static layers in single-AEM and single-CEM cells between IEM and cathode. The 'static' layers illustrated in the original **figure 1a** and **1b** were for the convenience of showing the movement of ions. To show the structure of the cells more clearly, we have modified **figure 1a** and **1b**, as shown on page 3 in the revised manuscript:

Figure 1. Schematic diagrams of alkaline polysulfide/air redox flow battery systems. a, RFB based on single anion exchange membrane. b, RFB based on single cation exchange membrane. c, RFB based on double membrane design combining anion exchange membrane and cation exchange membrane.

Therefore, we believe convection of intermediate solution has little effect on the ion transportation to the cathode and on the total performance of the battery because of following reasons: 1. In a dual-membrane structure the direction of convection in the intermediate solution is perpendicular to the hydroxide diffusion direction; 2. The convection is blocked by the AEM and will likely not affect the ion transport across it; 3. The rate-limiting process in this system is the cathode reaction, the impedance from ion transport contributes < 3% of the total impedance (figure S10).

As stated in question 1, we have assembled the single-CEM battery in liquid-gas-mixed-flowing state and its performance is inferior compared with the dual-membrane cell, which could be attributed to the blockage of oxygen transport pathways.

As oxygen only has about 1/10th the solubility in water as its concentration in air (~1 mM versus ~10 mM) and the diffusion coefficient of oxygen in water is about 10,000x slower ($10^{-5} \text{ cm}^2\text{s}^{-1}$ vs $0.1 \text{ cm}^2\text{s}^{-1}$) performance of the electrode, especially during discharge in the presence of liquid electrolyte is poor. When the cathode (air electrode) is immersed with liquid, the catalyst are submerged beneath a layer of water which will hamper oxygen transport to the catalytic sites, lowering performance, as has been well-investigated in the water management of proton-exchange membrane fuel cells^{16,17} (PEMFC). Hao et al.¹⁸ found that 10% water in the pore of an air electrode can start to deteriorate its performance. It has also been found that maintaining a suitable amount of water in the membrane to prevent membrane dehydration without flooding the cathode can best enhance air electrode performance. Due to this reason, it is common practise in fuel cell and air flow batteries to have direct contact between air electrode and the membrane without immersing it in liquid electrolyte. To the best of our knowledge, we could not find any work in the electrochemical energy field that uses gas-liquid mixed flow on the air side. Nevertheless, it might be possible to tune the hydrophobicity of the air electrode to design a cathode that can work with mixed-flow, but it involves detailed investigation of the air electrode and is beyond the scope of this work.

3. The Ref. 25 also used sodium salts (Na₂S₄, Na₂SO₄ and NaOH). So, the authors should also indicate sodium salts, not just lithium salts, when describing this work.

Ref. 25 indeed used sodium salt in the cost analysis and air electrode electrochemical test, but all their cell demonstrations (polarisation, cycling, power density etc. of polysulfide-air full cell) were performed using lithium salts and LiSICON membrane. To correctly relate the performance of different papers and generate the cost analysis, we chose to discuss Ref. 25 based on the actual materials they used to generate the performance data of the battery. More on this is elaborated in question 6.

To make the choice more clearly, a sentence is added to section *Cost analysis of flow battery system*, page 17:

“Reference work used sodium salt in the cost analysis and testing of air-electrode, and lithium salt in the actual polysulfide-air battery test. Therefore, to correctly correlate the battery performance with the cost of energy, lithium salt is chosen to calculate the energy cost of reference work.”

4. For Fig. 3, the authors compare the barrier performance of CEM and AEM on polysulfides. However, it can only explain that the cell structure in Fig. 1b is better than that in Fig. 1a, and cannot explain the advantages of adopting the cell structure in Fig. 1c.

We agree with the reviewer that the function of AEM should be stated more clearly. As answered in the first question, we add a polysulfide crossover experiment on the dual-membrane cell to investigate the effectiveness of dual-membrane structure in mitigating crossover. The experiment shows polysulfide crossover is slower in dual membrane structure than single CEM (**figure 3b, c and e** in the revised manuscript), suggesting that AEM has resistance to low-concentration polysulfide crossover.

This demonstrates the advantage of dual-membrane cell to single-CEM cell: In CEM cell, the cathode needs to be in flow state to avoid hydroxide starving in charging process, which brings a large difficulty in water management. With dual membrane structure, the cathode can be in gaseous state while polysulfide crossover is mitigated compared with single CEM cell. The result of the experiment is described in detail in page 9-10 of the revised manuscript.

5. In Fig. 4, as described above, when only CEM is used, the positive side is static, and the double membrane is used, the positive side is flow. The flow electrolyte has a great influence on the hydroxide diffusion, so, the results in Fig.4 cannot reflect the advantages of adding AEM. Why not use the same flow mode to evaluate the necessity of AEM? So, the comparison of electrochemical performance should be carried out with/without AEM using the same flow mode.

We thank the reviewer for this suggestion. As we have stated in the first and second question, we operated the single-CEM battery in mixed-flowing state. Although the polarisation during the charging process is lower due to sufficient OH⁻ supply, the polarisation during discharge is much higher than when using the dual membrane cell. This is because excess water has blocked the active sites when the cathode is operating in mixed-flow state. As already noted, it is common practise in fuel cell and air flow batteries to have direct contact between air electrode and the membrane without immersing it to liquid electrolyte because of this water-flooding problem. Therefore, in this work we think operating air electrodes in all our cells in air-flow states is more suitable for comparison between different cell structures and different publications.

6. In Fig. 6, the comparison of costs with Ref. 25 is hard to be persuasive. Two types of salts (Na/Li) are used in Ref. 25. If the price of Na salts is comparable with that of Li salts, it is obvious that the price of Na salts is cheaper than that of Li salts, and it has been reflected in Ref. 25 (Fig. 1). However, in Fig. 6b, the authors claim that the catholyte used in this work is cheaper than Ref. 25. Why? If it is all Na salts, what is the price difference between the two? The types of Na salts are different?

As stated in the *Cost analysis of flow battery system* section of the manuscript, the cost analysis in this work is based on the performance actually achieved in the work and the materials used to achieve that performance. Ref. 25 did use sodium salt in the work, however it is only used in the cost analysis and air electrode testing part. All the actual cell demonstrations (polarisation, cycling, power density etc. of polysulfide-air full cell) were performed using lithium salts and LiSICON solid-state membrane. Hence we find it strange that their cost analysis is based on sodium when they don't have any performance data for sodium. They explained that they chose the lithium salt because of the choice of LiSICON membrane, so we think choosing lithium salt is a choice relevant to the performance of the battery rather than a random choice. As different cations choice in membranes and salts can make a significant difference in the performance of electrochemical devices¹⁹⁻²¹, to correctly relate the performance of different papers and generate the cost analysis, we chose to calculate the cost of Ref. 25 based on the actual electrolyte they used to generate the performance data.

However to make our choice clearer, we have added a discussion on page 17:

“Reference work used sodium salt in the cost analysis and testing of air-electrode, and lithium salt in the actual polysulfide-air battery test. Therefore, to correctly correlate the battery performance with the cost of energy, lithium salt is chosen to calculate the energy cost of reference work.”

Reviewer #3 (Remarks to the Author):

This manuscript reports an experimental study of a dual-membrane PSA RFB, where a pair of AEM and CEM are installed. The device's configuration is effective and straightforward, but the implementation is incremental. Cross-over is significantly reduced, which makes the use of inexpensive catalysts possible. Relatively higher performance (than similar but single membrane setup) is obtained, although overall the power/energy densities, as well as lifetime (performance decay), are still much lower than incumbent technologies. The manuscript is well structured and clearly written. The design of this dual-membrane shows great potential for future development and application. Some comments about this manuscript are as follows.

1. Keep a reasonable number of figures and use only necessary images, e.g., Fig. 2 and Fig. 3. Some images are subjective, e.g., color change in Fig. 3. Also, cartoons in Fig. 3 are redundant and can be described in the text.

Thank you for the suggestions, we have merged the original **figure 3a and b**. We have also removed the original **figure 2d, h and i, figure 3g and h** and moved original **figure 2m** to **figure S2** in the revised SI. All the figures have been rearranged to make them clearer.

There is already a descriptive paragraph describing the ion transport in different cells on page 8 of the revised manuscript: *To conclude, FAA-3-PK-130 AEM would allow OH⁻ to pass through but is more vulnerable towards polysulfide crossover, while Nafion 117 CEM would allow Na⁺ to pass and have a higher resistance of polysulfide crossover but it would block OH⁻.*

The revised **figure 2** and **figure 3** are shown below:

Figure 2. Characterisation of electrodes. a, Diagram of sulfidized Ni foam electrode and MnO₂-based air electrode for OER and ORR reactions. b,c, SEM images of the commercial MnO₂-based air electrode. d, photo of the as-received Ni foam. e,f, SEM images of the as-received Ni foam. g, photo of the sulfidised Ni foam. h,i, SEM images of the sulfidised Ni foam. j,k, XPS core level spectra of the sulfidised Ni foam. l, polarisation measurements of the as-received and sulfidised Ni foams.

Figure 3. Diffusion of polysulfide species through ion-exchange membranes. a, Left: H-cell setup for polysulfide crossover determination on single AEM and CEM; Right: redox flow battery setup for setup for polysulfide crossover determination on dual membrane structure. b, Cumulative crossover of the permeate solution in single-AEM, single-CEM and dual-membrane setup against permeation time. c, Calculated polysulfide flux across single-AEM, single-CEM and dual-membrane structure. d, EIS of CEM and AEM. e, the polysulfide flux versus membrane conductivity.

2. It is felt that a cost analysis is somewhat premature for this topic since the performance is rather low and the decay rate is high. Efforts should be directed to gain an understanding of the mechanisms of degradation and possible approaches to enhance such design's performance and lifetime, as authors mentioned in lines 313-315 and lines 355-357.

The extremely low cost of energy storage is one of the main advantages of the polysulfide/air redox flow battery system, and is the reason it's so attractive despite the present low performance. To maintain the cost advantage of PSA RFB, it is important to enhance the electrochemical performance of the system while keeping the use of cost-effective materials (electrodes, membranes, electrolytes) to build the battery. Therefore, we think the costs (both of power and energy) are also important

indicators of its performance and should be presented in the publication along with other indicators such as peak power density, energy efficiencies, etc.

We agree that current effort should be put on gaining understanding on mechanisms and exploring approaches. We discuss the origin of degradation in page 16 of the revised manuscript. Further improvement of performance and stability will be presented in future works.

3. Fig. 4e and Fig. 5e show voltage variation vs. capacity for different cycles. It is curious that the results of the 10th cycle and 20th cycle appear to be very close to each other. Please provide an explanation and discussion.

Figure 4e, figure 5d and figure 5e in the original manuscript represents the potential of the full-cell, the polysulfide-side half-cell and the air-side half-cell respectively, during the 30 charge and discharge cycles. It can be seen from **figure 5 d, e** that the polysulfide-side half-cell potential decreased from the first cycle to a minimum at the 10th cycle and then increased slightly at the 20th cycle and decreased again at the 30th cycle, whereas the air-side half-cell potential decreased steadily along the cycle number. Since the full-cell potential is the difference between the positive and negative half-cell potentials, the overall results exhibited a similar full-cell potential at the 10th and the 20th cycle. The slight potential increase of the polysulfide-side half-cell at the 20th cycle compared to the 10th cycle could be attributed to the activity improvement of the negative electrode over the first few cycles since nickel can be sulfidised during the oxidation the polysulfide electrolyte and nickel sulfide is considered as a better catalyst for the polysulfide redox reaction.

Similar phenomenon is found in the new 80-cycle cycling experiments, the degradation of the cell slowed down between the 10-30th cycle and then accelerated after 30th cycle, which is more obvious on the charging plateau. The degradation of air electrode is more significant than the polysulfide electrode, suggesting the above phenomenon is common in PSA RFB system. Our future research will focus on improving the activity and stability of the air electrode.

3. The tested current densities in this study are low, thus diffusion of species appears to be the predominating mechanism of transport. Authors might also want to consider other mechanisms such as electro-osmotic drag, in particular on how these transport mechanisms may affect the (limiting) air-side electrode. Please elaborate.

As the reviewer pointed out, the transport of species should be mainly through diffusion. Convection is normally an important transport mechanism in redox flow batteries, because of the flowing state of electrolyte. In our work we believe it is less significant because the following three reasons: 1. The oxygen side is separated from intermediate flowing solution; 2. The flowing direction of intermediate solution is perpendicular to the hydroxide transportation direction; 3. The current density is relatively small.

The electro-osmotic effect would influence the water management of air-electrode^{16,22}, which consequently influences the hydroxide diffusion through the AEM.

During the charging process, water and oxygen is produced at the air electrode, and OH⁻ moves from polysulfide side to air side simultaneously. Electro-osmotic drag causes the water to move from intermediate solution to the air electrode, causing the air electrode to be flooded. However, the influence of water flooding to the oxygen evolution reaction at air side during charging process is minimal as the main reactant is OH⁻ from the aqueous phase, and the oxygen will spontaneously exit the cathode.

During the discharging process, the water is consumed at the air side and OH⁻ moves from the air electrode to the polysulfide side. The electro-osmotic drag will further cause the water to move in the

same direction as the OH⁻, i.e. from air side to polysulfide side. This potentially causes AEM drying out and lowering of its hydroxide conductivity. Therefore, during the operation we bubbled the oxygen through hydroxide solution so the oxygen flow could keep the cathode humidified. We didn't observe any reduction of performance caused by the AEM drying out.

The above discussion is added to revised SI on page 10-11:

- 1 Jeong, S., Kim, L.-H., Kwon, Y. & Kim, S. Effect of nafion membrane thickness on performance of vanadium redox flow battery. *Korean J. Chem. Eng.* **31**, 2081-2087, doi:10.1007/s11814-014-0157-5 (2014).
- 2 Wang, J. *et al.* Poly(aryl piperidinium) membranes and ionomers for hydroxide exchange membrane fuel cells. *Nature Energy* **4**, 392-398, doi:10.1038/s41560-019-0372-8 (2019).
- 3 Guo, W. *et al.* Strategies and insights towards the intrinsic capacitive properties of MnO₂ for supercapacitors: Challenges and perspectives. *Nano Energy* **57**, 459-472, doi:<https://doi.org/10.1016/j.nanoen.2018.12.015> (2019).
- 4 Li, Z. *et al.* Air-Breathing Aqueous Sulfur Flow Battery for Ultralow-Cost Long-Duration Electrical Storage. *Joule* **1**, 306-327, doi:<https://doi.org/10.1016/j.joule.2017.08.007> (2017).
- 5 Zhang, S. *et al.* Recent Progress in Polysulfide Redox-Flow Batteries. *Batteries & Supercaps* **2**, 627-637, doi:<https://doi.org/10.1002/batt.201900056> (2019).
- 6 Allen, P. L. & Hickling, A. Electrochemistry of sulphur. Part 1.—Overpotential in the discharge of the sulphide ion. *Transactions of the Faraday Society* **53**, 1626-1635, doi:10.1039/TF9575301626 (1957).
- 7 Lessner, P., McLarnon, F. R., Winnick, J. & Cairns, E. J. Kinetics of Aqueous Polysulfide Solutions: Part III . Investigation of Homogeneous and Electrode Kinetics by the Rotating Disk Method. *J. Electrochem. Soc.* **134**, 2669-2677, doi:10.1149/1.2100270 (1987).
- 8 Stenina, I. A., Sístat, P., Rebrov, A. I., Pourcelly, G. & Yaroslavtsev, A. B. Ion mobility in Nafion-117 membranes. *Desalination* **170**, 49-57, doi:<https://doi.org/10.1016/j.desal.2004.02.092> (2004).
- 9 Lehmani, A., Turq, P., Périé, M., Périé, J. & Simonin, J.-P. Ion transport in Nafion[®] 117 membrane. *Journal of Electroanalytical Chemistry* **428**, 81-89, doi:[https://doi.org/10.1016/S0022-0728\(96\)05060-7](https://doi.org/10.1016/S0022-0728(96)05060-7) (1997).
- 10 Sijabat, R. R., de Groot, M. T. & van der Schaaf, J. Maxwell-Stefan modeling and experimental study on the ionic resistance of cation-selective membranes in concentrated lye solutions. *Journal of Membrane Science* **607**, 118134, doi:<https://doi.org/10.1016/j.memsci.2020.118134> (2020).
- 11 Hagemann, T. *et al.* An aqueous all-organic redox-flow battery employing a (2,2,6,6-tetramethylpiperidin-1-yl)oxyl-containing polymer as catholyte and dimethyl viologen dichloride as anolyte. *J. Power Sources* **378**, 546-554, doi:<https://doi.org/10.1016/j.jpowsour.2017.09.007> (2018).
- 12 Yan, L. *et al.* Balancing Osmotic Pressure of Electrolytes for Nanoporous Membrane Vanadium Redox Flow Battery with a Draw Solute. *ACS Appl. Mater. Interfaces* **8**, 35289-35297, doi:10.1021/acsami.6b12068 (2016).
- 13 Xia, Y. *Development and characterisation of anode materials for polysulphide-air redox flow battery*, Imperial College London, (2019).
- 14 Hirsch, A. R. & Zavala, G. Long-term effects on the olfactory system of exposure to hydrogen sulphide. *Occup. Environ. Med.* **56**, 284-287, doi:10.1136/oem.56.4.284 (1999).
- 15 Baran, M. J. *et al.* Design Rules for Membranes from Polymers of Intrinsic Microporosity for Crossover-free Aqueous Electrochemical Devices. *Joule* **3**, 2968-2985, doi:<https://doi.org/10.1016/j.joule.2019.08.025> (2019).
- 16 Moçotéguy, P., Ludwig, B., Beretta, D. & Pedersen, T. Study of the impact of water management on the performance of PEMFC commercial stacks by impedance spectroscopy.

- Int. J. Hydrogen Energy* **45**, 16724-16737, doi:<https://doi.org/10.1016/j.ijhydene.2020.04.139> (2020).
- 17 Yousfi-Steiner, N. *et al.* A review on PEM voltage degradation associated with water management: Impacts, influent factors and characterization. *J. Power Sources* **183**, 260-274, doi:<https://doi.org/10.1016/j.jpowsour.2008.04.037> (2008).
- 18 Hao, L., Moriyama, K., Gu, W. & Wang, C.-Y. Three Dimensional Computations and Experimental Comparisons for a Large-Scale Proton Exchange Membrane Fuel Cell. *J. Electrochem. Soc.* **163**, F744-F751, doi:10.1149/2.1461607jes (2016).
- 19 Symington, A. R. *et al.* Quantifying the impact of disorder on Li-ion and Na-ion transport in perovskite titanate solid electrolytes for solid-state batteries. *J. Mat. Chem. A* **8**, 19603-19611, doi:10.1039/D0TA05343K (2020).
- 20 Jin, W., Du, H., Zheng, S., Xu, H. & Zhang, Y. Comparison of the Oxygen Reduction Reaction between NaOH and KOH Solutions on a Pt Electrode: The Electrolyte-Dependent Effect. *The Journal of Physical Chemistry B* **114**, 6542-6548, doi:10.1021/jp102367u (2010).
- 21 Kacprzak, A., Kobylecki, R. & Bis, Z. Influence of temperature and composition of NaOH–KOH and NaOH–LiOH electrolytes on the performance of a direct carbon fuel cell. *J. Power Sources* **239**, 409-414, doi:<https://doi.org/10.1016/j.jpowsour.2013.03.159> (2013).
- 22 Eriksson, B. *et al.* Quantifying water transport in anion exchange membrane fuel cells. *Int. J. Hydrogen Energy* **44**, 4930-4939, doi:<https://doi.org/10.1016/j.ijhydene.2018.12.185> (2019).

REVIEWER COMMENTS

Reviewer #1 (Remarks to the Author):

This manuscript is recommended for publication since the authors responded thoroughly to my comments.

Minor:

1) Figure 3a: I would recommend that the authors include a schematic of the polysulfide diffusivity test setup as inset or in the Supporting Information, which will provide better information than photos.

2) Figure 4f: Although the authors briefly mentioned that the catalytic activity loss (the gradual dissolution of MnO₂) could be the main cause of the rapid decay in RFB performance (i.e., VE, EE) on page 15. I would strongly recommend that the authors should explain it in more detail with appropriate citations.

Reviewer #2 (Remarks to the Author):

All previous questions have been well addressed, it is now recommended for acceptance.

We are grateful to reviewer 1 for additional comments on our manuscript. We have addressed the comments and made necessary change to the revised manuscript, which is marked in red.

Below we list the detailed comments (black) and our point-to-point response (blue).

REVIEWER COMMENTS

Reviewer #1 (Remarks to the Author):

This manuscript is recommended for publication since the authors responded thoroughly to my comments.

Minor:

1) Figure 3a: I would recommend that the authors include a schematic of the polysulfide diffusivity test setup as inset or in the Supporting Information, which will provide better information than photos.

Thank you for your suggestion. We replaced the photos in **Figure 3 (a)** with schematics of diffusivity experiment setups. The photos of the setups are now placed in **Figure S6**, in page 6 of the supporting information. The figure numbers were adjusted accordingly. The figure is referred to in page 7 of the revised manuscript.

Figure 3. Diffusion of polysulfide species through ion-exchange membranes. *a*, Left: H-cell setup for polysulfide crossover determination on single AEM (FAA-3-PK-130) and CEM (Nafion 117); Right: redox flow battery setup for polysulfide crossover determination on dual membrane structure. *b*, Cumulative crossover of the permeate solution in single-AEM, single-CEM and dual-membrane setup against permeation time. *c*, Calculated polysulfide flux across single-AEM, single-CEM and dual-membrane structure. *d*, EIS of CEM and AEM. *e*, the polysulfide flux versus membrane conductivity.

Figure S6. Photos of polysulfide crossover determination setups: (a) H-cell setup for single-membrane structure and for (b) redox-flow battery setup for dual-membrane structure

2) Figure 4f: Although the authors briefly mentioned that the catalytic activity loss (the gradual dissolution of MnO_2) could be the main cause of the rapid decay in RFB performance (i.e., VE, EE) on page 15. I would strongly recommend that the authors should explain it in more detail with appropriate citations.

As we mentioned in page 11, in single-CEM battery, the water used to flush the air electrode after 20 cycles turned purple. We also found similar phenomenon in dual-membrane battery after 80 cycles. We have found literatures investigating $\text{MnO}_4^- / \text{MnO}_2$ redox couple in both acidic¹ and alkaline^{2,3} conditions. The standard potential of $\text{MnO}_4^- / \text{MnO}_2$ is 1.7 V vs. SHE in acidic condition and 0.56 V vs. SHE in alkaline condition. The cathode potential in our dual-membrane polysulfide-air RFB system during charge is close to 0.56 V (**Figure 5e** and **Figure S12**) so MnO_2 dissolution is a likely parasitic reaction.

The kinetic of MnO_2 electrochemical dissolution should be fairly slow since it did not significantly influence the battery's performance until after 200 h of cycle. This is also evidenced by the absence of

report of MnO₂ dissolution in all literatures we read applying MnO₂ as OER catalyst in alkaline condition⁴⁻⁷. Even though the voltage applied in their OER tests were well above 0.56 V vs. SHE.

Nonetheless, it is necessary to use a more stable air catalyst to operate the polysulfide-air RFB system in a larger timescale.

A discussion on the electrode degradation and MnO₂ dissolution is added in page 15:

“To conclude, the degradation of this alkaline PSA RFB was mainly attributed to the catalytic activity loss of the air electrode towards OER, which is likely due to the dissolution of MnO₂. The MnO₄⁻ / MnO₂ redox couple has been investigated in both acidic and alkaline conditions. Its standard potential in alkaline condition is 0.56 V vs. SHE, which is close to the half-cell voltage of air electrode during charge. Therefore, the oxidation of MnO₂ to highly dissolvable MnO₄⁻ is deduced to be the parasitic reaction that causes the air electrode degradation.”

- 1 Yadav, G. G., Turney, D., Huang, J., Wei, X. & Banerjee, S. Breaking the 2 V Barrier in Aqueous Zinc Chemistry: Creating 2.45 and 2.8 V MnO₂-Zn Aqueous Batteries. *ACS Energy Letters* **4**, 2144-2146, doi:10.1021/acsenergylett.9b01643 (2019).
- 2 Licht, S., Ghosh, S., Naschitz, V., Halperin, N. & Halperin, L. Fe(VI) Catalyzed Manganese Redox Chemistry: Permanganate and Super-Iron Alkaline Batteries. *The Journal of Physical Chemistry B* **105**, 11933-11936, doi:10.1021/jp012178t (2001).
- 3 San-Chung, L., Yung-Yun, W., Chi-Chao, W. & -Cheng, C. J. Reinvestigation of the Electrochemical Reduction of KMnO₄. *Bull. Chem. Soc. Jpn.* **66**, 3372-3376, doi:10.1246/bcsj.66.3372 (1993).
- 4 Bera, K. *et al.* Enhancement of the OER Kinetics of the Less-Explored α-MnO₂ via Nickel Doping Approaches in Alkaline Medium. *Inorg. Chem.* **60**, 19429-19439, doi:10.1021/acs.inorgchem.1c03236 (2021).
- 5 Meng, Y. *et al.* Structure–Property Relationship of Bifunctional MnO₂ Nanostructures: Highly Efficient, Ultra-Stable Electrochemical Water Oxidation and Oxygen Reduction Reaction Catalysts Identified in Alkaline Media. *J. Am. Chem. Soc.* **136**, 11452-11464, doi:10.1021/ja505186m (2014).
- 6 Cheng, F., Su, Y., Liang, J., Tao, Z. & Chen, J. MnO₂-Based Nanostructures as Catalysts for Electrochemical Oxygen Reduction in Alkaline Media. *Chem. Mater.* **22**, 898-905, doi:10.1021/cm901698s (2010).
- 7 Morgan Chan, Z. *et al.* Electrochemical trapping of metastable Mn³⁺ ions for activation of MnO₂; oxygen evolution catalysts. *Proceedings of the National Academy of Sciences* **115**, E5261, doi:10.1073/pnas.1722235115 (2018).